# Understanding Aggregations of Proper Learners in Multiclass Classification

**Julian Asilis**                                                          ASILIS@USC.EDU
*University of Southern California*

**Mikael Møller Høgsgaard**                                    HOGSGAARD@CS.AU.DK
*Aarhus University*

**Grigoris Velegkas**                                   GRIGORIS.VELEGKAS@YALE.EDU
*Yale University*

**Editors:** Gautam Kamath and Po-Ling Loh

## Abstract

Multiclass learnability is known to exhibit a properness barrier: there are learnable classes which cannot be learned by any proper learner. Binary classification faces no such barrier for learnability, but a similar one for *optimal* learning, which can in general only be achieved by improper learners. Fortunately, recent advances in binary classification have demonstrated that this requirement can be satisfied using aggregations of proper learners, some of which are strikingly simple. This raises a natural question: to what extent can simple aggregations of proper learners overcome the properness barrier in multiclass classification?

We give a positive answer to this question for classes which have finite Graph dimension, $d_G$. Namely, we demonstrate that the optimal binary learners of Hanneke, Larsen, and Aden-Ali et al. (appropriately generalized to the multiclass setting) achieve sample complexity $O\left(\frac{d_G + \ln(1/\delta)}{\epsilon}\right)$. This forms a strict improvement upon the sample complexity of ERM. We complement this with a lower bound demonstrating that for certain classes of Graph dimension $d_G$, majorities of ERM learners require $\Omega\left(\frac{d_G + \ln(1/\delta)}{\epsilon}\right)$ samples. Furthermore, we show that a single ERM requires $\Omega\left(\frac{d_G \ln(1/\epsilon) + \ln(1/\delta)}{\epsilon}\right)$ samples on such classes, exceeding the lower bound of Daniely et al. (2015) by a factor of $\ln(1/\epsilon)$. For multiclass learning in full generality — i.e., for classes of finite DS dimension but possibly infinite Graph dimension — we give a strong refutation to these learning strategies, by exhibiting a learnable class which cannot be learned to constant error by *any* aggregation of a finite number of proper learners.

**Keywords:** Multiclass classification, proper learning, ERM, majority voting.

## 1. Introduction

Multiclass classification, the task of learning to classify data into different categories, is one of the archetypal problems of machine learning (ML), rich with real-world applications. Perhaps the best-known such application is image recognition, the task of learning a mapping from images — represented by a matrix of pixels — to their respective classes, whose number may be very large. On the practical side, a landmark result of Krizhevsky et al. (2012) illustrated the capabilities of neural networks on this problem and sparked the modern neural-network-based ML revolution, which has recently culminated in the advent of large language models.

Despite impressive progress from practitioners, however, our theoretical understanding of multiclass classification remains relatively limited. In fact, even for the simple case of *binary* classification — in which there are only two possible labels for the data — the problem of designing

an optimal learner in Valiant's *Probably Approximately Correct* (PAC) framework (Valiant, 1984) was only recently resolved by a breakthrough result of Hanneke (2016). In the (realizable) PAC model, there is an underlying data-generating process, modeled as a distribution $\mathcal{D}$, a hypothesis class $\mathcal{H}$ which contains a *perfect* classifier $h^*$, and a learner which uses i.i.d. samples drawn from $\mathcal{D}$ in order to output a classifier. The goal of the learner is to emit a classifier with a low probability of misclassifying a fresh training point drawn from $\mathcal{D}$. Notably, the fundamental theorem of statistical learning theory asserts that the simple principle of Empirical Risk Minimization (ERM), defined as simply outputting any of the hypotheses in $\mathcal{H}$ with best performance on the training sample, is nearly-optimal. Hanneke (2016), however, improved upon the performance of ERM in binary classification by designing an algorithm with provably smaller error.

Roughly speaking, Hanneke's algorithm operates by building $\approx n^{0.79}$ sub-samples from the training set $S$ (where $|S| = n$), calling ERM on each such sub-sample, and outputting the majority vote of these classifiers. Notably, the sub-samples are designed to leave out certain portions of the original sample, and a sophisticated inductive argument which leverages this structure is used to prove optimality. More recently, Larsen (2023) demonstrated that this sub-sampling strategy may be replaced with the simpler technique of drawing samples from $S$ with replacement, and furthermore establishes optimality using only $O(\log(n/\delta))$ sub-samples. Strikingly, this strategy of drawing from $S$ with replacement, calling ERM on each such sub-sample, and combining the resulting classifiers via majority vote is precisely the celebrated classical technique of *bagging*, or bootstrap aggregation, introduced by Breiman (1996). Yet another advancement in optimal binary classification was recently made by Aden-Ali et al. (2024), who demonstrated that a majority of just three ERMs trained on disjoint $1/3$ fractions of the training set also achieves optimality *in expectation*, as well as in the high-probability regime for a certain range of parameters. (We will formalize the distinction between the high-probability and in-expectation models shortly.)

It is worth remarking that these three optimal learners share a great deal of structure: they act by drawing sub-samples of the training set $S$, calling an ERM learner on each subsample, and aggregating the resulting classifiers with a majority vote. Furthermore, ERM is an example of a *proper* learner, i.e., a learner which always emits a classifier that lies in the underlying hypothesis class $\mathcal{H}$. It has been demonstrated that proper learners are in general unable to achieve optimal error rates in binary classification (Bousquet et al., 2020), and it thus may come as a surprise that an aggregation of a small number of proper learners (only 3!) is able to achieve optimality.

Let us return now to the general multiclass classification setting, in which the number of possible labels is permitted to be arbitrarily large, even infinite. To what extent does the landscape of learning differ from the binary case? Perhaps more than one may expect. A long and beautiful line of work has demonstrated that the multiclass setting exhibits several fundamental differences from that of binary classification (Natarajan and Tadepalli, 1988; Natarajan, 1989b; Ben-David et al., 1992; Haussler and Long, 1995; Rubinstein et al., 2006; Daniely et al., 2012; Daniely and Shalev-Shwartz, 2014; Daniely et al., 2015; Brukhim et al., 2022). In particular, it was demonstrated by Daniely and Shalev-Shwartz (2014) that there exist learnable multiclass problems which can only be learned by improper learners, i.e., those which may emit hypotheses outside of $\mathcal{H}$. Consequently, ERM fails on some learnable multiclass classification problems, in contrast to the binary case. Daniely et al. (2015) showed that a hypothesis class is learnable by the ERM rule precisely when its *graph dimension* is finite. (Finiteness of the graph dimension, then, is a sufficient but not necessary condition for learnability.) The problem of characterizing *improper* multiclass learnability via a dimension remained a foremost open question for several decades, and was recently resolved by the break-

through work of Brukhim et al. (2022). There, they demonstrated that learnability is characterized by the finiteness of the *Daniely-Shwartz* (DS) dimension (Daniely and Shalev-Shwartz, 2014), and provided a learning strategy that achieves learnability when this dimension is finite. At its heart, their algorithm is based upon a careful extension of the *one-inclusion graph* (OIG) predictor of Haussler et al. (1994) to the multiclass setting, and introduces several beautiful theoretical ideas, such as *list* PAC learning. Interestingly, the algorithmic approach of Brukhim et al. (2022) is strikingly different from — and more complex than — ERM and related algorithmic approaches from binary classification.

In light of this recent breakthrough in multiclass classification, along with the recent advances in optimal binary classification, it is natural to ask: Can the algorithmic approaches that lead to optimal learners for binary classification problems be extended to the multiclass setting? More precisely, we ask:

> *When do simple aggregations of proper learners perform well in multiclass classification? Can any such learner succeed on all multiclass problems possible?*

## 1.1. Results

Our main results give a fairly complete understanding to the previous question and can be summarized as follows:

- **Upper bound on majorities of ERMs for Graph classes.** Our first set of main results concerns majorities of ERMs for classes with finite Graph dimension, following the natural generalization of the algorithms from Hanneke (2016); Larsen (2023); Aden-Ali et al. (2024) to the multiclass setting. For both Hanneke's algorithm and bagging, we show that for confidence parameter $\delta$ and error rate $\varepsilon$, the number of training samples needed to achieve this guarantee is $O\left(\frac{d_G + \ln(1/\delta)}{\varepsilon}\right)$ (cf. Theorem 5), where $d_G$ denotes the Graph dimension of the class. Recall that Daniely et al. (2012) had shown that ERM needs at most $O\left(\frac{d_G \ln(1/\varepsilon) + \ln(1/\delta)}{\varepsilon}\right)$ samples, so this result shaves off a logarithmic factor in the upper bound. Subsequently, we also show that the majority of three ERMs algorithm of Aden-Ali et al. (2024) requires at most $O\left(\frac{d_G}{\varepsilon}\right)$ number of samples to obtain error $\varepsilon$, in expectation.

- **Lower bound on ERMs and majorities of ERMs.** It is natural to ask whether the upper bounds we establish are tight, or whether they may be room for improvement in their analysis. Theorem 10 demonstrates that for any $d_G$, there indeed exists a class with Graph dimension $d_G$ and constant DS dimension (hence learnable), for which taking majorities over ERMs requires at least $\Omega\left(\frac{d_G + \ln(1/\delta)}{\varepsilon}\right)$ samples to achieve error $\varepsilon$ with confidence $\delta$. Moreover, we also show that using simple ERMs for this class requires at least $\Omega\left(\frac{d_G \ln(1/\varepsilon) + \ln(1/\delta)}{\varepsilon}\right)$ samples, exceeding the lower bound of Daniely et al. (2015, Theorem 9) by a factor of $\ln(1/\varepsilon)$. (Notably, however, their lower bound holds for *all* classes of finite Graph dimension, whereas ours holds for a particular family of classes.)

- **Lower bound on combinations of proper learners.** Given our negative result about majorities of ERMs, it is natural to ask whether *arbitrary combinations* of *arbitrary proper learners*

could lead to learning algorithms for all classes with finite DS dimension. Theorem 13 provides a strong negative result: there exists a hypothesis class of DS dimension 1 for which any learner achieving error below $1/2$ cannot be expressed as *any* aggregation of finitely many proper learners.

### 1.2. Techniques

We now give a high-level overview of the techniques we use to establish our results. For the upper bound on majorities of ERMs, our analysis begins by relating the multiclass hypothesis class $\mathcal{H}$ to an induced binary hypothesis class $\bar{\mathcal{H}}$ whose VC dimension is bounded by the Graph dimension of the original class. This connection was also utilized by Daniely et al. (2015) and can be traced back to Natarajan (1989a); Ben-David et al. (1995). Subsequently, we show that the sample complexities of the algorithms that are used to learn $\mathcal{H}$ can be upper bounded by their sample complexities for learning $\bar{\mathcal{H}}$. Our argument makes use of the particular structure of these algorithms, i.e., that the manner in which they generate sub-training sequences is oblivious to the datapoints' values, instead using only their indices. See Section 3.1 for detail, including the *index-splitting* property of Definition 4. One interesting aspect of our approach is that in the analysis of the upper bound, we only count the prediction as being correct if a $1/2$-majority of the learners predicts correctly. (E.g., any test point on which the predictions do not reach a majority vote is automatically treated as a misclassification.) Our matching lower bound illustrates that this seemingly wasteful analysis is in fact tight.

Moving on to the sample complexity lower bound for ERMs and their majorities, our construction is inspired by the first Cantor class lower bound of Daniely and Shalev-Shwartz (2014), combined with a coupon collector argument of Auer and Ortner (2007). The latter is what allows us to get an improved lower bound for ERMs compared to that of Daniely and Shalev-Shwartz (2014). See Section 3.2 for detail. It is worth highlighting that we can immediately get lower bounds for a family of classes with DS dimension equal to their Graph dimension, by trivially extending the binary classification lower bounds to the multiclass caser. However, in our construction, the family that witnesses the lower bound has *constant* DS dimension, hence every class is learnable with $\tilde{O}(1/\varepsilon)$ many samples (Brukhim et al., 2022).

Lastly, we move on to the lower bound regarding arbitrary aggregations of proper learners. By carefully modifying the *first Cantor classs* of Daniely and Shalev-Shwartz (2014) and Daniely et al. (2015), we exhibit a class of DS dimension which cannot be learned to constant error by any aggregation of a finite number of proper learners — regardless of the aggregation strategy or the complexities of the constituent proper learners. Furthermore, our results are articulated using *properness numbers*, which measure the "extent" to which a classifier $f \colon \mathcal{X} \to \mathcal{Y}$ or learner $\mathcal{A}$ is improper, and may be of independent interest to the community.

### 1.3. Related Work

**Binary Classification.** The pioneering work of Valiant (1984) proposed the PAC learning framework for binary classification, which has since been perhaps the most well-studied notion of learning. Subsequently, Blumer et al. (1989a) showed that a combinatorial measure previously introduced by Vapnik and Chervonenkis (1971) precisely captures learnability in this setting and that ERM is an (almost) optimal learner. Building upon the work of Simon (2015), Hanneke (2016) designed the first optimal PAC learner for binary classification, based on running ERMs on different

carefuly chosen sub-samples of the training data, thereby resolving a decades-long open problem. Two more optimal learners have since been introduced; Aden-Ali et al. (2023) designed an optimal PAC learner based on the one-inclusion graph predictor of Haussler et al. (1994), and Larsen (2023) proved that the well-known bagging algorithm of Breiman (1996) is also optimal. Very recently, Aden-Ali et al. (2024) showed that taking a majority vote of three ERMs trained on non-overlapping sub-samples of the training data is optimal *in expectation.*

**Multiclass Classification.** Moving on to the general multiclass classification setting, early attempts characterize learnability focused on the case where the number of labels is *finite*. In this setting, Natarajan and Tadepalli (1988), Natarajan (1989b) and Ben-David et al. (1995) identified natural generalization of the VC dimension, such as the *Natarajan* dimension, whose finiteness characterizes learnability. For the setting of infinitely many labels, it long remained an open problem whether finiteness of the Natarajan dimension continues to characterize learnablity. The breakthrough result of Brukhim et al. (2022) demonstrated that this is not the case, and that learnability is instead characterized by the finiteness of the *DS dimension* (Daniely and Shalev-Shwartz, 2014). Surprisingly, Daniely and Shalev-Shwartz (2014) showed that optimal learners for multiclass classification cannot in general be *proper*, i.e., they may have to output hypotheses outside of the underlying class. Regarding strategies for optimal multiclass learning, Daniely and Shalev-Shwartz (2014) demonstrated that certain orientations of one-inclusion graphs are optimal in the *transductive* model of learning, and nearly-optimal in the high-probability regime. (Detailing the transductive model is slightly beyond the scope of this paper; see Vapnik and Chervonenkis (1974), Vapnik (1982), or Haussler et al. (1994) for some of its seminal work.) More recently, Asilis et al. (2024b) demonstrated that multiclass classification problems always admit optimal learners governed by a relaxed form of regularization which they term *unsupervised local regularization*, and conjectured that their results can be improved to hold for *local regularization* (Asilis et al., 2024a). Neither form of learner can be seen as an aggregation of proper learners, however, and thus does not directly address our primary question.

## 2. Preliminaries

### 2.1. Notation

For a natural number $n \in \mathbf{N}$, we let $[n] := \{1, \dots, n\}$. We will denote random variables with bold-faced letters (e.g., $\mathbf{x}$) and realizations of them using non-bold typeface (e.g., $x$). For a set $Z$, we let $Z^*$ denote the set of all finite sequences in $Z$, i.e., $Z^* = \bigcup_{i \in \mathbb{N}} Z^i$. We define $\overrightarrow{\mathbf{1}}$ as the all-ones vector, whose length will always be clear from context. When $A$ and $B$ are sequences, we use $A \oplus B$ to denote their concatenation. If $f \colon \mathcal{X} \to \mathcal{Y}$ is a function and $X \subseteq \mathcal{X}$ a subset, $f|_X \colon X \to \mathcal{Y}$ denotes the restriction of $f$ to $X$. Likewise, if $\mathcal{F} \subseteq \mathcal{Y}^\mathcal{X}$ is a collection of functions, $\mathcal{F}|_X = \{f|_X : f \in \mathcal{F}\}$ denotes the set of $\mathcal{F}$'s restrictions to $X$.

### 2.2. Learning Theory

We let $\mathcal{X}$ denote the **feature space** and $\mathcal{Y}$ the **label space**. An element of $(\mathcal{X} \times \mathcal{Y})$ is referred to as an **example**. In accordance with the above, we let $(\mathcal{X} \times \mathcal{Y})^*$ denote the set of all possible **training sequences**. For a training sequence $S \in (\mathcal{X} \times \mathcal{Y})^*$, we sometimes write $S = (S_\mathcal{X}, S_\mathcal{Y})$ for the part of the training examples in $\mathcal{X}, \mathcal{Y}$, respectively, i.e., such that $S_\mathcal{X} \in \mathcal{X}^*$ and $S_\mathcal{Y} \in \mathcal{Y}^*$. For a training sequence $S$, we say that $S'$ is a **sub-training sequence** of $S$ if $(x, y) \in S'$ implies that $(x, y) \in S$.

Note that this does *not* imply that $S'$ has the same multiplicity of a given training example $(x, y)$ as $S$ does; we will in fact permit $S'$ to contain examples with greater multiplicity than $S$.

We refer to a function $f\colon \mathcal{X} \to \mathcal{Y}$ as a **hypothesis** or *classifier*, and to a collection of such functions $\mathcal{H} \subseteq \mathcal{Y}^{\mathcal{X}}$ as a **hypothesis class**. For a distribution $\mathcal{D}$ over $\mathcal{X} \times \mathcal{Y}$ and a **hypothesis** $h \in \mathcal{Y}^{\mathcal{X}}$, we write $\mathcal{L}_{\mathcal{D}}(h) = \mathbb{E}_{(\mathbf{x}, \mathbf{y}) \sim \mathcal{D}}[h(\mathbf{x}) \neq y]$ for the **error**, or *true error*, of $h$ under $\mathcal{D}$. For a distribution $\mathcal{D}$ over the feature space $\mathcal{X}$ and a classifier $c$, we let $\mathcal{D}_c$ denote the distribution over $\mathcal{X} \times \mathcal{Y}$ which draws unlabeled data according to $\mathcal{D}$ and labels it using $c$. That is, $\mathcal{D}_c(A) = \mathbb{P}_{\mathbf{x} \sim \mathcal{D}}((\mathbf{x}, c(\mathbf{x})) \in A)$.

We say that a hypothesis $h$ is **consistent** with a training sequence $S \in (\mathcal{X} \times \mathcal{Y})^*$ if for all $(x, y) \in S$ we have that $h(x) = y$. A training sequence $S$ is said to be **realizable** by $\mathcal{H}$ if there exists an $h \in \mathcal{H}$ which is consistent with $S$. Similarly, a distribution $\mathcal{D}$ over $\mathcal{X} \times \mathcal{Y}$ is *realizable* by a class $\mathcal{H}$ if for all training sequences $S$ in the support of $\mathbf{S} \sim \mathcal{D}^m$, we have that $S$ is realizable by $\mathcal{H}$. A **learner** is a function from training sequences to classifiers, e.g., $\mathcal{A} : (\mathcal{X} \times \mathcal{Y})^* \to \mathcal{Y}^{\mathcal{X}}$. Such a learner is said to be **proper** with respect to an underlying hypothesis class $\mathcal{H}$ if the image of $\mathcal{A}$ lies in $\mathcal{H}$, meaning $\mathcal{A}(S) \in \mathcal{H}$ for all training sequences $S$. Otherwise, $\mathcal{A}$ is **improper**. If for all realizable training sequence $S$, $\mathcal{A}$ has the property of emitting a hypothesis in $\mathcal{H}$ which is consistent with $S$, then $\mathcal{A}$ is an **ERM learner** for $\mathcal{H}$.

Let the label $\bot$ denote the act of deliberately making an error when performing classification over any label space $\mathcal{Y}$. (Semantically, this can be thought of as forfeiting one's prediction at a feature point, or as an "I don't know" label.) For a set of classifiers $f_1, \ldots, f_n \in \mathcal{Y}^{\mathcal{X}}$, we define their **majority** $\mathrm{Maj}(f_1, \ldots, f_n)$ as follows:

$$
\mathrm{Maj}(f_1, \ldots, f_n)(x) = \begin{cases} y & \text{if } \exists y \in \mathcal{Y} \text{ s.t. } \forall y' \neq y, \\ & |\{f_i : f_i(x) = y\}| > |\{f_i : f_i(x) = y'\}|; \\ \hat{y} \in \{\bot, f_1(x), \ldots, f_n(x)\} & \text{otherwise.} \end{cases}
$$

In short, $\mathrm{Maj}(f_1, \ldots, f_n)$ outputs the strictly most popular label for the point $x$ if such a label exists, and otherwise is permitted to arbitrarily agree with a classifier $f_i$ or to emit the $\bot$ label. (Strictly speaking, it would be more accurate to refer to refer to it as a *plurality* classifier, but we retain language from the binary case for the sake of simplicity.) In some cases we use $\mathrm{Maj}_{\bot}$ to denote the majority voter which always predicts $\bot$ in the latter case, a kind of worst-case majority voter. Notably, all our results — both lower- and upper-bounds — hold for *any* majority learner following the above the structure. Thus, our flexibility in defining majority voting has the effect of strengthening our results, relative to a more narrow definition.

We now recall Valiant's celebrated Probably Approximately Correct (PAC) learning model, also known as the high-probability model of learning (Valiant, 1984).

**Definition 1** *A learner $\mathcal{A}$ is said to be a **PAC learner** for a hypothesis class $\mathcal{H} \subseteq \mathcal{Y}^{\mathcal{X}}$ if there exists a function $m\colon (0, 1)^2 \to \mathbf{N}$ with the following property: for any distribution $\mathcal{D}$ over $\mathcal{X} \times \mathcal{Y}$ which is realizable with respect to $\mathcal{H}$, and for any $\epsilon, \delta \in (0, 1)$, when $\mathcal{A}$ is trained on a sample $S$ of $m(\epsilon, \delta)$ many points drawn i.i.d. from $\mathcal{D}$, then*

$$
\mathcal{L}_{\mathcal{D}}\left(\mathcal{A}(S)\right) \leq \epsilon
$$

*with probability at least $1 - \delta$ over the random draw of $S$.*

We note that the minimal function $m$ for which Definition 1 holds is referred to as the **sample complexity** of the learner $\mathcal{A}$. Similarly, the *sample complexity* of a hypothesis class $\mathcal{H}$ is the minimal

sample complexity attained by any of its PAC learners. Any learner attaining this sample complexity — up to a multiplicative constant — is said to be **optimal** for $\mathcal{H}$ in the PAC model (also referred to as the high-probability model). The expected error model, meanwhile, is that in which one instead requires $\mathbb{E}_S \left[ \mathcal{L}_\mathcal{D} \mathcal{A}(S) \right] \leq \epsilon$, and considers the sample complexity of achieving this only as a function of $\epsilon$, the error parameter.

Many of our results reference the *Graph dimension* of a hypothesis class.

**Definition 2** *A hypothesis class* $\mathcal{H} \subseteq \mathcal{Y}^\mathcal{X}$ **Graph shatters** *a set of points* $(x_1, y_1), \ldots, (x_d, y_d) \in (\mathcal{X} \times \mathcal{Y})$ *if for all* $b \in \{0, 1\}^d$, *there exists* $h \in \mathcal{H}$ *such that* $h(x_i) = y_i$ *if* $b_i = 1$ *and* $h(x_i) \neq y_i$ *if* $b_i = 0$. *The* **Graph dimension** *of* $\mathcal{H}$, *denoted* $d_G = d_G(\mathcal{H})$, *is the size of the largest set shattered by* $\mathcal{H}$ *(or* $\infty$ *if arbitrarily large sets are shattered).*

Note that for the case of binary classification, in which $|\mathcal{Y}| = 2$, the Graph dimension equals precisely the VC dimension, for any hypothesis class. It will also be useful to define the DS dimension (Daniely and Shalev-Shwartz, 2014; Brukhim et al., 2022).

**Definition 3** *A hypothesis class* $\mathcal{H} \subseteq \mathcal{Y}^\mathcal{X}$ **DS shatters** *a set of points* $(x_1, \ldots, x_d) \in \mathcal{X}$ *if there exists a finite non-empty subset* $\mathcal{F} \subseteq \mathcal{H}$ *such that for each* $h \in \mathcal{F}$ *and* $i \in [d]$, *there exists a* $g \in \mathcal{F}$ *such that* $g(i) \neq h(i)$ *and* $g(j) = h(j)$ *for all* $j \neq i$. *Such a* $g$ *is referred to as an* $i$-neighbor *of* $h$. *The* **DS dimension** *of* $\mathcal{H}$, *denoted* $d_{DS} = d_{DS}(\mathcal{H})$, *is the size of the largest set DS shattered by* $\mathcal{H}$ *(or* $\infty$ *if arbitrarily large sets are shattered).*

We briefly remark that we assume standard measurability assumptions on $\mathcal{X}$ and $\mathcal{H}$ throughout the remainder of the paper; see e.g. Shalev-Shwartz and Ben-David (2014), Blumer et al. (1989b), or (Pollard, 2012, Appendix C).

## 3. Learning with Simple Majorities (of ERM)

This section is devoted to our results — both positive and negative — for ERM learners and their majorities in the multiclass setting. We commence with positive results in Section 3.1 and proceed to negative results in Section 3.2.

### 3.1. Upper Bound

We now demonstrate how one can translate upper bounds on majority voters in the binary case to the multiclass case. To this end, we draw inspiration from the multiclass to binary classification reduction as it appears in (Daniely et al., 2015, p. 6), which is due to Natarajan (1989a) and Ben-David et al. (1995). We begin by describing the central ideas and definitions of the reduction.

For a hypothesis $h \in \mathcal{H}$, define $\bar{h}$ as the mapping from $\mathcal{X} \times \mathcal{Y}$ to $\{0, 1\}$, which on input $(x_1, x_2) \in \mathcal{X} \times \mathcal{Y}$ outputs $\bar{h}(x_1, x_2) = \mathbb{1}\{h(x_1) = x_2\}$. Furthermore, let $\bar{\mathcal{H}} = \{\bar{h}\}_{h \in \mathcal{H}}$. We first claim that $\text{VC}(\bar{\mathcal{H}}) = d_G(\mathcal{H})$. To see this, let $(x_1, y_1), \ldots, (x_d, y_d)$ be a sequence shattered by $\bar{\mathcal{H}}$, meaning that for each $b \in \{0, 1\}^d$, there exists $\bar{h} \in \bar{\mathcal{H}}$ with $b_i = \bar{h}(x_i, y_i) = \mathbb{1}\{h(x_i) = y_i\}$. Then for all $b \in \{0, 1\}^d$, there exists a $h \in \mathcal{H}$ such that $h(x_i) = y_i$ if $b_i = 1$ and $h(x_i) \neq y_i$ if $b_i = 0$. Thus $(x_1, y_1), \ldots, (x_d, y_d)$ is Graph shattered by $\mathcal{H}$, as desired. Conversely, if $(x_1, y_1), \ldots, (x_d, y_d)$ is Graph shattered by $\mathcal{H}$, then for every $b \in \{0, 1\}^d$ there exists a $h \in \mathcal{H}$ such that $h(x_i) = y_i$ if and only if $b_i = 1$. Then $b_i = \mathbb{1}\{h(x_i) = y_i\} = \bar{h}(x_i, y_i)$, and thus each $b \in \{0, 1\}^d$ can be witnessed as a behavior of $\bar{\mathcal{H}}$ on $(x_1, \ldots, x_d)$, completing the argument.

Note that if a distribution $\mathcal{D}$ over $\mathcal{X} \times \mathcal{Y}$ is realizable by $\mathcal{H}$, then the distribution $\mathcal{D}_1$ over $\mathcal{X} \times \mathcal{Y} \times \{1\}$ defined by $\mathcal{D}_1(A \times \{1\}) = \mathcal{D}(A)$ is likewise realizable by $\bar{\mathcal{H}}$. This follows essentially immediately from the definition of $\bar{h}$ as the characteristic function of the graph of $h$. For a learner $\mathcal{A}$ for $\mathcal{H}$, define the learner $\bar{\mathcal{A}}$ for $\bar{\mathcal{H}}$ that on input $S = (S_\mathcal{X}, S_\mathcal{Y}, S_{\{0,1\}}) \in (\mathcal{X} \times \mathcal{Y} \times \{0,1\})^*$ outputs $\overline{\mathcal{A}(S_\mathcal{X}, S_\mathcal{Y})}$. (I.e., $\bar{\mathcal{A}}$ does not use the information of $S_{\{0,1\}}$.) We note that if a distribution $\mathcal{D}$ is realizable by $\mathcal{H}$ and $\mathcal{A}$ is an ERM-learner for $\mathcal{H}$, then $\bar{\mathcal{A}}$ is also an ERM-learner for $\mathcal{D}_1$ and $\bar{\mathcal{H}}$. In particular, for a training sequence $\mathbf{S} = (\mathbf{S}_\mathcal{X}, \mathbf{S}_\mathcal{Y}, \mathbf{S}_{\{0,1\}}) \sim \mathcal{D}_1$, we have that $\bar{\mathcal{A}}(\mathbf{S}) \in \bar{\mathcal{H}}$ and furthermore for any $(x, y, b) \in \mathbf{S}$,

$$\bar{\mathcal{A}}(\mathbf{S}_\mathcal{X}, \mathbf{S}_\mathcal{Y}, \mathbf{S}_{\{0,1\}})(x, y) = \mathbb{1}\{\mathcal{A}(\mathbf{S}_\mathcal{X}, \mathbf{S}_\mathcal{Y})(x) = y\} = 1 = b,$$

where the final equality uses the definition of $\mathcal{D}_1$. We now define a *splitting algorithm*, which takes as input a single training sequence and outputs a family of training sequences. The following definition captures the bagging routine used in Larsen (2023), the deterministic splitting scheme of Hanneke (2016), and the very simple splitting scheme used by Aden-Ali et al. (2024) which splits the input training sequence into 3 disjoint equal-sized parts.

**Definition 4** *We say that $\mathcal{S}$ is a **splitting algorithm** if, for any input space $\mathcal{X}$, label space $\mathcal{Y}$, and training sequence $S \in (\mathcal{X} \times \mathcal{Y})^*$, the output of $\mathcal{S}$ on $S$ is a family of training sequences $\mathcal{S}(S) = (S^1, S^2 \dots, S^m)$ such that every $S^i$ is a sub-training sequence of $S$. Further, each such splitting algorithm must satisfy the **index splitting property**: For any input sequence $S$, the sub-training sequences $S^1, \dots, S^m$ are constructed so as to only depend upon the indices of elements in $S$, not their values.*[1]

Intuitively, the index splitting property requires that the splitting algorithm generate the subtraining sequences by only using the indices in $S$, and is oblivious to the information within the training examples (i.e., their features and labels). Thus, a splitting algorithm is essentially independent of the input space $\mathcal{X}$ and label space $\mathcal{Y}$. We will permit splitting algorithms to be randomized, in which case we write them as $\mathcal{S}_r$, where $r$ denotes the source of randomness (e.g., a random binary string). We note that randomization is employed by Larsen (2023), where each $S^i$ is a sequence drawn with replacement from $S$ (and $r$ controls the randomness of the draw). To adapt Definition 4, we would require that the index splitting property hold for each realization of $r$.

We will often consider one sequence of examples in $\mathcal{X} \times \mathcal{Y}$ and another sequence of examples in $\mathcal{X} \times \mathcal{Y} \times \{1\}$, as outlined at the beginning of the section. We may denote the former as $(S_\mathcal{X}, S_\mathcal{Y})$ and the latter as $(S_\mathcal{X}, S_\mathcal{Y}, \overrightarrow{\mathbf{1}})$, with $S_\mathcal{X} \in \mathcal{X}^*$, $S_\mathcal{Y} \in \mathcal{Y}^*$, and $\overrightarrow{\mathbf{1}}$ the all-ones vector. Note that the index splitting property implies that the number of sub-training sequences in $\mathcal{S}_r(S_\mathcal{X}, S_\mathcal{Y}, \overrightarrow{\mathbf{1}})$ is the same as $\mathcal{S}_r(S_1, S_\mathcal{Y})$, i.e. $|\mathcal{S}_r(S_\mathcal{X}, S_\mathcal{Y}, \overrightarrow{\mathbf{1}})| = |\mathcal{S}_r(S_1, S_\mathcal{Y})|$. We further note that if $S' = (S_1', S_2') \in \mathcal{S}_r(S)$ is realizable by $\mathcal{H}$ then $(S_1', S_2', \overrightarrow{\mathbf{1}}) \in \mathcal{S}_r((S, \overrightarrow{\mathbf{1}}))$ is realizable by $\bar{\mathcal{H}}$. We now present the splitting schemes of Hanneke (2016), Larsen (2023), and Aden-Ali et al. (2024), for which we will derive upper bounds over the course of this section.

The splitting scheme of Hanneke (2016) is deterministic; as such, we denote it by $\mathcal{S}^H$, omitting dependence upon any random string $r$. With a slight abuse of notation, we define a function of *two*

---

1. More precisely, a splitting algorithm $\mathcal{S}$ with the index splitting property amounts to a choice of value $\mathcal{S}_n \in ([n]^*)^*$ for each $n \in \mathbb{N}$. Letting $S = (s_1, \dots, s_n)$ be a sequence of length $n$ and denoting $\mathcal{S}_n = \left((i_1^1, \dots, i_{m_1}^1), \dots, (i_1^k, \dots, i_{m_k}^k)\right)$, one then defines $\mathcal{S}(S) = \left((s_{i_1^1}, \dots, s_{i_{m_1}^1}), \dots, (s_{i_1^k}, \dots, s_{i_{m_k}^k})\right)$.

arguments $\mathcal{S}^H(S, T)$ in Algorithm 1. Hanneke's splitting algorithm is then defined as $\mathcal{S}^H(S) = \mathcal{S}^H(S, \emptyset)$. The splitting algorithm has a recursive structure, as we now describe. (Recall that $A \oplus B$ denotes the concatenation of sequences $A$ and $B$.)

---

**Algorithm 1** $\mathcal{S}^H(S, T)$

---

1: **Input:** Training sequences $S, T \in (\mathcal{X}, \mathcal{Y})^*$
2: **Output:** Family of sub training sequences of $S \oplus T$
3: **If** $|S| \leq 3$
4:      **return** $S \oplus T$
5: Let $S_0$ denote the first $|S| - 3\lfloor|S|/4\rfloor$ elements of $S$
6: Let $S_1, S_2, S_3$ denote the first, second and third next $\lfloor|S|/4\rfloor$ elements of $S$, respectively, following the elements $S_0$
7: **return** $[\mathcal{S}^H(S_0, S_2 \oplus S_3 \oplus T), \mathcal{S}^H(S_0, S_1 \oplus S_3 \oplus T), \mathcal{S}^H(S_0, S_1 \oplus S_2 \oplus T)]$

---

We now define the splitting scheme of Larsen (2023), which — in contrast to Hanneke's method — employs randomness. Furthermore, it takes as input a failure parameter $\delta$ and a bagging size parameter $\rho \in [0.02, 1]$. We denote this bagging splitting scheme by $\mathcal{S}_r^B$, and emphasize that the randomness $r \sim \mathbf{r}$ it employs is independent of $S$, in accordance with the index splitting property.

---

**Algorithm 2** $\mathcal{S}_r^B(S, T)$

---

1: **Input:** Training sequence $S \in (\mathcal{X}, \mathcal{Y})^*$, bagging size parameter $\rho$ and failure probability $\delta$
2: **Output:** Family of sub training sequences of $S$
3: **For** $i$ in $[\lceil 18 \ln(2|S|/\delta)\rceil]$
4:      Let $S_i$ denote a sample of size $\rho \cdot |S|$ drawn with replacement from $S$
5: **return** $[S_1, \ldots, S_{\lceil 18 \ln(2|S|/\delta)\rceil}]$

---

Lastly, the splitting algorithm of Aden-Ali et al. (2024) is deterministic; we denote it $\mathcal{S}^T$.

---

**Algorithm 3** $\mathcal{S}^T(S)$

---

1: **Input:** Training sequence $S \in (\mathcal{X}, \mathcal{Y})^*$
2: **Output:** Family of sub training sequences of $S$
3: Let $S_1, S_2, S_3$ denote the first, second and third sequence of $\lfloor|S|/3\rfloor$ elements, respectively
4: **return** $[S_1, S_2, S_3]$

---

With the above algorithms introduced, we are now equipped to present the main theorem of this section, which gives generalization bounds for the above splitting schemes in the multiclass setting.

**Theorem 5** *Let $a_B, a_H, a_T$ denote universal constants. Let $\mathcal{H} \subseteq \mathcal{Y}^\mathcal{X}$ be a hypothesis class, $\mathcal{D}$ a distribution over $\mathcal{X} \times \mathcal{Y}$ which is realizable with respect to $\mathcal{H}$, and $\mathcal{A}$ an ERM-learner for $\mathcal{H}$. Let $\mathcal{H}$ have Graph dimension $d_G$, and let $\mathcal{S}^H, \mathcal{S}_r^B, \mathcal{S}^T$ be the previously defined splitting algorithms. Then each of the following hold.*

- *For any $\delta \in (0, 1)$, with probability at least $1 - \delta$ over $\mathbf{S} \sim \mathcal{D}^m$ one has that*

$$\mathcal{L}_\mathcal{D}\left(\mathrm{Maj}\left(\mathcal{A}(\mathcal{S}^H(\mathbf{S}))\right)\right) \leq a_H \frac{d_G + \ln(1/\delta)}{m}.$$

- *For any $\delta \in (0, 1)$ and bagging size parameter $\rho \in [0.02, 1]$, with probability at least $1 - \delta$ over $\mathbf{S} \sim \mathcal{D}^m$ and the randomness $\mathbf{r}$ employed in bagging, one has that*

$$\mathcal{L}_\mathcal{D}\left(\mathrm{Maj}\left(\mathcal{A}(\mathcal{S}^B_{\mathbf{r}}(\mathbf{S}))\right)\right) \leq a_B \frac{d_G + \ln(1/\delta)}{m}.$$

- *The expected error incurred by $\mathcal{A}$'s majority vote over the splitting algorithm $\mathcal{S}^T$ is (at most) linear in each of $d_G$ and $m^{-1}$, i.e.,*

$$\mathop{\mathbb{E}}_{\mathbf{S} \sim \mathcal{D}^m}\left[\mathcal{L}_\mathcal{D}\left(\mathrm{Maj}\left(\mathcal{A}(\mathcal{S}^T(\mathbf{S}))\right)\right)\right] \leq a_T \frac{d_G}{m}.$$

We now move on to prove the above theorem. To this end, we require the following lemma, which establishes that the error of the majority vote of a learner $\mathcal{A}$ can be upper bounded by that of $\bar{\mathcal{A}}$. Recall that when $\mathcal{D}$ is a distribution over $\mathcal{X} \times \mathcal{Y}$, $\mathcal{D}_1$ denotes the distribution over $\mathcal{X} \times \mathcal{Y} \times \{1\}$ with $\mathcal{D}_1(A \times \{1\}) = \mathcal{D}(A)$.

**Lemma 6** *Let $\mathcal{H} \subseteq \mathcal{Y}^\mathcal{X}$ be a hypothesis class and $\mathcal{D}$ a realizable distribution over $\mathcal{X} \times \mathcal{Y}$. Further, let $\mathcal{A}$ be a learner for $\mathcal{H}$ and let $S_r$ be a (possibly randomized) splitting algorithm. Then for any possible source of randomness $r \in \mathcal{R}$, we have*

$$\mathop{\mathbb{E}}_{\mathbf{S} \sim \mathcal{D}^m}\left[\mathcal{L}_\mathcal{D}\left(\mathrm{Maj}(\mathcal{A}(\mathcal{S}_r(\mathbf{S})))\right)\right] \leq \mathop{\mathbb{E}}_{\mathbf{S} \sim \mathcal{D}_1^m}\left[\mathcal{L}_{\mathcal{D}_1}\left(\mathrm{Maj}_\perp(\bar{\mathcal{A}}(\mathcal{S}_r(\mathbf{S})))\right)\right],$$

*and moreover for any $\epsilon > 0$,*

$$\mathop{\mathbb{P}}_{\mathbf{S} \sim \mathcal{D}^m}\left[\mathcal{L}_\mathcal{D}\left(\mathrm{Maj}(\mathcal{A}(\mathcal{S}_r(\mathbf{S})))\right) > \epsilon\right] \leq \mathop{\mathbb{P}}_{\mathbf{S} \sim \mathcal{D}_1^m}\left[\mathcal{L}_{\mathcal{D}_1}\left(\mathrm{Maj}_\perp(\bar{\mathcal{A}}(\mathcal{S}_r(\mathbf{S})))\right) > \epsilon\right].$$

The above lemma upper bounds the error of a majority vote in the multiclass setting by that of its induced binary classifier. We postpone its proof for the moment, as it is slightly involved, and proceed with the proof of Theorem 5. Let us recall the guarantees of Hanneke (2016), Larsen (2023) and Aden-Ali et al. (2024) for optimal binary classification, which we will employ in the course of generalizing their results to the multiclass case (for classes of finite Graph dimension).

**Theorem 7 (Hanneke (2016, Theorem 2))** *There is a universal constant $a_H > 0$ such that for every binary hypothesis class $\bar{\mathcal{H}}$, realizable distribution $\mathcal{D}$, ERM-learner $\bar{\mathcal{A}}$, and value $\delta \in (0, 1)$, then with probability at least $1 - \delta$ over the choice of training set $\mathbf{S} \sim \mathcal{D}^m$, one has*

$$\mathcal{L}_{\mathcal{D}_c}\left(\mathrm{Maj}_\perp\left(\bar{\mathcal{A}}(\mathcal{S}^H(\mathbf{S}))\right)\right) \leq a_H \frac{d + \ln(1/\delta)}{m},$$

*where $d = \mathrm{VC}(\bar{\mathcal{H}})$ denotes the VC dimension of $\bar{\mathcal{H}}$.*

**Theorem 8 (Larsen (2023, Theorem 1))** *There is a universal constant $a_B > 0$ such that for every binary hypothesis class $\bar{\mathcal{H}}$, realizable distribution $\mathcal{D}$, ERM-learner $\bar{\mathcal{A}}$, value $\delta \in (0,1)$, and bagging size parameter $\rho \in [0.02, 1]$, then with probability at least $1 - \delta$ over the random choice of training set $\mathbf{S} \sim \mathcal{D}^m$ and random string $\mathbf{r}$ used to generate the bootstrap training sequences,*

$$\mathcal{L}_{\mathcal{D}_c}\left(\mathrm{Maj}_\perp\left(\bar{\mathcal{A}}(\mathcal{S}_\mathbf{r}^B(\mathbf{S})))\right)\right) \leq a_B \frac{d + \ln(1/\delta)}{m},$$

*where $d = \mathrm{VC}(\bar{\mathcal{H}})$ denotes the VC dimension of $\bar{\mathcal{H}}$.*

**Theorem 9 (Aden-Ali et al. (2024, Theorem 1.1))** *There is a universal constant $a_T > 0$ such that for every binary hypothesis class $\bar{\mathcal{H}}$, realizable distribution $\mathcal{D}$, and ERM-learner $\bar{\mathcal{A}}$,*

$$\mathbb{E}_{\mathbf{S}\sim\mathcal{D}^m}\left[\mathcal{L}_{\mathcal{D}_c}\left(\mathrm{Maj}_\perp\left(\bar{\mathcal{A}}(\mathcal{S}_r^T(\mathbf{S})))\right)\right)\right] \leq a_T \frac{d + \ln(1/\delta)}{m},$$

*where $d = \mathrm{VC}(\bar{\mathcal{H}})$ denotes the VC dimension of $\bar{\mathcal{H}}$.*

With the above theorems in place, we are now equipped to prove Theorem 5.

**Proof of Theorem 5** Recall from the beginning of the section that for any hypothesis class $\mathcal{H} \subseteq \mathcal{Y}^\mathcal{X}$ in multiclass classification, the Graph dimension $d_G$ of $\mathcal{H}$ is equal to the VC dimension of $\bar{\mathcal{H}}$, where $\bar{\mathcal{H}} = \{\bar{h} : h \in \mathcal{H}\} \subseteq \{0,1\}^{\mathcal{X}\times\mathcal{Y}}$ and $\bar{h}(x,y) = \mathbb{1}\{h(x) = y\}$. Further, we established that when $\mathcal{D}$ is an $\mathcal{H}$-realizable distribution over $\mathcal{X} \times \mathcal{Y}$ and $\mathcal{D}_1$ is the distribution over $\mathcal{X} \times \mathcal{Y} \times \{1\}$ defined by $\mathcal{D}_1(A \times \{1\}) = \mathcal{D}(A)$, then $\mathcal{D}_1$ is likewise $\bar{\mathcal{H}}$-realizable. Lastly, recall that if $\mathcal{A}$ is an ERM-learner for $\mathcal{H}$, then $\bar{\mathcal{A}}$ is an ERM-learner for $\bar{\mathcal{H}}$ on samples drawn from any distribution of the form $\{\mathcal{D}_1 : \mathcal{D}$ is $\mathcal{H}$-realizable$\}$, where $\bar{\mathcal{A}}(S_\mathcal{X}, S_\mathcal{Y}, S_{\{0,1\}}) = \overline{\mathcal{A}(S_\mathcal{X}, S_\mathcal{Y})}$.

We now prove the claim of the theorem for $\mathcal{S}_\mathbf{r}^B$; the other results follow in a similar fashion. Let $r$ be any realization of the randomness of $\mathbf{r}$. By invoking Theorem 6 with $\mathcal{S}_r^B$ and $\varepsilon = a_B(d + \ln(1/\delta))/m$, we have that

$$\mathbb{P}_{\mathbf{S}\sim\mathcal{D}^m}\left[\mathcal{L}_\mathcal{D}\left(\mathrm{Maj}(\mathcal{A}(\mathcal{S}_r^B(\mathbf{S})))\right) > \varepsilon\right] \leq \mathbb{P}_{\mathbf{S}\sim\mathcal{D}_1^m}\left[\mathcal{L}_{\mathcal{D}_1}\left(\mathrm{Maj}_\perp(\bar{\mathcal{A}}(\mathcal{S}_r^B(\mathbf{S})))\right) > a_B \frac{d + \ln(1/\delta)}{m}\right].$$

As this holds for any realization $r$ of $\mathbf{r}$, and $\mathbf{r}$ and $\mathbf{S}$ are independent, we have by Theorem 8 that

$$\mathbb{P}_{\substack{\mathbf{S}\sim\mathcal{D}^m, \\ \mathbf{r}}}\left[\mathcal{L}_\mathcal{D}\left(\mathrm{Maj}(\mathcal{A}(\mathcal{S}_\mathbf{r}^B(\mathbf{S})))\right) > \varepsilon\right] \leq \mathbb{P}_{\substack{\mathbf{S}\sim\mathcal{D}_1^m, \\ \mathbf{r}}}\left[\mathcal{L}_{\mathcal{D}_1}\left(\mathrm{Maj}_\perp(\bar{\mathcal{A}}(\mathcal{S}_\mathbf{r}^B(\mathbf{S})))\right) > a_B \frac{d + \ln(1/\delta)}{m}\right] \leq \delta.$$

$\blacksquare$

It remains only to complete the proof of Theorem 6, with which we conclude the section.

**Proof of Theorem 6** Let $S = (S_\mathcal{X}, S_\mathcal{Y}) \in (\mathcal{X} \times \mathcal{Y})^*$. Since we have for $(x, y) \in \mathcal{X} \times \mathcal{Y}$ that $\bar{\mathcal{A}}(S_\mathcal{X}, S_\mathcal{Y}, \overrightarrow{\mathbf{1}})(x, y) = \mathbb{1}\{\mathcal{A}(S_\mathcal{X}, S_\mathcal{Y})(x) = y\}$, we conclude that $\mathbb{1}\{\mathcal{A}(S)(x) = y\} = \mathbb{1}\{\bar{\mathcal{A}}(S_\mathcal{X}, S_\mathcal{Y}, \overrightarrow{\mathbf{1}})(x, y) = 1\}$. Thus, we have that

$$\sum_{S^* \in \mathcal{S}_r(S)} \frac{\mathbb{1}\{\mathcal{A}(S^*)(x) = y\}}{|\mathcal{S}_r(S)|} = \sum_{S^* \in \mathcal{S}_r(S)} \frac{\mathbb{1}\{\bar{\mathcal{A}}(S_\mathcal{X}^*, S_\mathcal{Y}^*, \overrightarrow{\mathbf{1}})(x, y) = 1\}}{|\mathcal{S}_r(S)|}.$$

By the index splitting property, the sum over $S^* = (S_{\mathcal{X}}^*, S_{\mathcal{Y}}^*)$ in $\mathcal{S}_r(S) = \mathcal{S}_r(S_{\mathcal{X}}, S_{\mathcal{Y}})$ can be written as the sum over over $S^* = (S_{\mathcal{X}}^*, S_{\mathcal{Y}}^*, \overrightarrow{\mathbf{1}})$ in $\mathcal{S}_r(S_{\mathcal{X}}, S_{\mathcal{Y}}, \overrightarrow{\mathbf{1}})$. Thus,

$$\sum_{S^* \in \mathcal{S}_r(S)} \frac{\mathbb{1}\{\mathcal{A}(S^*)(x) = y\}}{|\mathcal{S}_r(S)|} = \sum_{S^* \in \mathcal{S}_r(S_{\mathcal{X}}, S_{\mathcal{Y}}, \overrightarrow{\mathbf{1}})} \frac{\mathbb{1}\{\bar{\mathcal{A}}(S_{\mathcal{X}}^*, S_{\mathcal{Y}}^*, \overrightarrow{\mathbf{1}})(x, y) = 1\}}{|\mathcal{S}_r(S_{\mathcal{X}}, S_{\mathcal{Y}}, \overrightarrow{\mathbf{1}})|}.$$

As the above holds for each $(x, y) \in \mathcal{X} \times \mathcal{Y}$, we have that

$$\mathbb{P}_{(\mathbf{x},\mathbf{y})\sim\mathcal{D}}\left[\sum_{\mathbf{S}^* \in \mathcal{S}_r(S)} \frac{\mathbb{1}\{\mathcal{A}(S^*)(\mathbf{x}) = \mathbf{y}\}}{|\mathcal{S}_r(S)|} \leq 1/2\right] = \mathbb{P}_{(\mathbf{x},\mathbf{y})\sim\mathcal{D}}\left[\sum_{\mathbf{S}^* \in \mathcal{S}_r(S_{\mathcal{X}}, S_{\mathcal{Y}}, \overrightarrow{\mathbf{1}})} \frac{\mathbb{1}\{\bar{\mathcal{A}}(S_{\mathcal{X}}^*, S_{\mathcal{Y}}^*, \overrightarrow{\mathbf{1}})(\mathbf{x}, \mathbf{y}) = 1\}}{|\mathcal{S}_r(S_{\mathcal{X}}, S_{\mathcal{Y}}, \overrightarrow{\mathbf{1}})|} \leq 1/2\right]$$

$$= \mathbb{P}_{(\mathbf{x},\mathbf{y},\mathbf{1})\sim\mathcal{D}_1}\left[\sum_{\mathbf{S}^* \in \mathcal{S}_r(S_{\mathcal{X}}, S_{\mathcal{Y}})} \frac{\mathbb{1}\{\bar{\mathcal{A}}(S_{\mathcal{X}}^*, S_{\mathcal{Y}}^*, \overrightarrow{\mathbf{1}})(\mathbf{x}, \mathbf{y}) = 1\}}{|\mathcal{S}_r(S_{\mathcal{X}}, S_{\mathcal{Y}}, \overrightarrow{\mathbf{1}})|} \leq 1/2\right].$$

Now, let $\mathrm{Maj}(\mathcal{A}(\mathcal{S}(S)))$ denote the majority vote of the classifiers $\{\mathcal{A}(S^*)\}_{S^* \in \mathcal{S}(S)}$. Then, by definition of the majority vote, the event $\mathrm{Maj}(\mathcal{A}(\mathcal{S}(S)))(x) \neq y$ is disjoint from the event that $\sum_{S^* \in \mathcal{S}_r(S)} \mathbb{1}\{\mathcal{A}(S^*)(x) = y\} > 1/2 \cdot |\mathcal{S}_r(S)|$ and thus must be a subset of its complement, i.e., the event that $\sum_{\mathbf{S}^* \in \mathcal{S}_r(S)} \mathbb{1}\{\mathcal{A}(S^*)(x) = y\} \leq 1/2|\mathcal{S}_r(S)|$. Using the above, we have that

$$\mathbb{P}_{(\mathbf{x},\mathbf{y})\sim\mathcal{D}}\left[\mathrm{Maj}(\mathcal{A}(\mathcal{S}(S)))(\mathbf{x}) \neq \mathbf{y}\right] = \mathbb{P}_{(\mathbf{x},\mathbf{y})\sim\mathcal{D}}\left[\mathrm{Maj}(\mathcal{A}(\mathcal{S}(S)))(\mathbf{x}) \neq \mathbf{y}, \sum_{\mathbf{S}^* \in \mathcal{S}_r(S)} \frac{\mathbb{1}\{\mathcal{A}(S^*)(\mathbf{x}) = \mathbf{y}\}}{|\mathcal{S}_r(S)|} \leq 1/2\right]$$

$$+ \mathbb{P}_{(\mathbf{x},\mathbf{y})\sim\mathcal{D}}\left[\mathrm{Maj}(\mathcal{A}(\mathcal{S}(S)))(\mathbf{x}) \neq \mathbf{y}, \sum_{\mathbf{S}^* \in \mathcal{S}_r(S)} \frac{\mathbb{1}\{\mathcal{A}(S^*)(\mathbf{x}) = \mathbf{y}\}}{|\mathcal{S}_r(S)|} > 1/2\right]$$

$$\leq \mathbb{P}_{(\mathbf{x},\mathbf{y})\sim\mathcal{D}}\left[\sum_{\mathbf{S}^* \in \mathcal{S}_r(S)} \frac{\mathbb{1}\{\mathcal{A}(S^*)(\mathbf{x}) = \mathbf{y}\}}{|\mathcal{S}_r(S)|} \leq 1/2\right]$$

$$= \mathbb{P}_{(\mathbf{x},\mathbf{y},\mathbf{1})\sim\mathcal{D}_1}\left[\sum_{\mathbf{S}^* \in \mathcal{S}_r(S_{\mathcal{X}}, S_{\mathcal{Y}}, \overrightarrow{\mathbf{1}})} \frac{\mathbb{1}\{\bar{\mathcal{A}}(S_{\mathcal{X}}^*, S_{\mathcal{Y}}^*, \overrightarrow{\mathbf{1}})(\mathbf{x}, \mathbf{y}) = 1\}}{|\mathcal{S}_r(S_{\mathcal{X}}, S_{\mathcal{Y}}, \overrightarrow{\mathbf{1}})|} \leq 1/2\right].$$

As this holds for each choice of $S \in (\mathcal{X} \times \mathcal{Y})^*$, we further have that

$$\mathbb{E}_{\mathbf{S}\sim\mathcal{D}^m}\left[\mathbb{P}_{(\mathbf{x},\mathbf{y})\sim\mathcal{D}}[\mathrm{Maj}(\mathcal{A}(\mathcal{S}(\mathbf{S})))(\mathbf{x}) \neq \mathbf{y}]\right]$$

$$\leq \mathbb{E}_{\mathbf{S}\sim\mathcal{D}^m}\left[\mathbb{P}_{(\mathbf{x},\mathbf{y},\mathbf{1})\sim\mathcal{D}_1}\left[\sum_{\mathbf{S}^* \in \mathcal{S}_r(\mathbf{S}_{\mathcal{X}}, \mathbf{S}_{\mathcal{Y}}, \overrightarrow{\mathbf{1}})} \frac{\mathbb{1}\{\bar{\mathcal{A}}(\mathbf{S}_{\mathcal{X}}^*, \mathbf{S}_{\mathcal{Y}}^*, \overrightarrow{\mathbf{1}})(\mathbf{x}, \mathbf{y}) = 1\}}{|\mathcal{S}_r(\mathbf{S}_{\mathcal{X}}, \mathbf{S}_{\mathcal{Y}}, \overrightarrow{\mathbf{1}})|} \leq 1/2\right]\right]$$

$$= \mathbb{E}_{\mathbf{S}\sim\mathcal{D}_1^m}\left[\mathbb{P}_{(\mathbf{x},\mathbf{y},\mathbf{1})\sim\mathcal{D}_1}\left[\sum_{\mathbf{S}^* \in \mathcal{S}_r(\mathbf{S})} \frac{\mathbb{1}\{\bar{\mathcal{A}}(\mathbf{S}^*)(\mathbf{x}, \mathbf{y}) = 1\}}{|\mathcal{S}_r(\mathbf{S})|} \leq 1/2\right]\right]$$

$$\leq \mathbb{E}_{\mathbf{S}\sim\mathcal{D}_1^m}\left[\mathcal{L}_{\mathcal{D}_1}\left(\mathrm{Maj}_\perp(\mathcal{A}(\mathcal{S}_r(\mathbf{S})))\right)\right],$$

where we recall that the $\mathrm{Maj}_\perp$ denotes the decision rule which outputs $\perp$ (i.e., deliberately chooses to fail) when there is no strictly most-popular label. Then for any $\epsilon > 0$,

$$
\mathop{\mathbb{P}}_{\mathbf{S} \sim \mathcal{D}^m} \left[ \mathop{\mathbb{P}}_{(\mathbf{x},\mathbf{y}) \sim \mathcal{D}} [\mathrm{Maj}(\mathcal{A}(\mathcal{S}(\mathbf{S})))(\mathbf{x}) \neq \mathbf{y}] > \epsilon \right]
$$

$$
\leq \mathop{\mathbb{P}}_{\mathbf{S} \sim \mathcal{D}^m} \left[ \mathop{\mathbb{P}}_{(\mathbf{x},\mathbf{y},\mathbf{1}) \sim \mathcal{D}_1} \left[ \sum_{\mathbf{S}^* \in \mathcal{S}_r(\mathbf{S}_\mathcal{X}, \mathbf{S}_\mathcal{Y}, \overrightarrow{\mathbf{1}})} \frac{\mathbb{1}\{\bar{\mathcal{A}}(\mathbf{S}_\mathcal{X}^*, \mathbf{S}_\mathcal{Y}^*, \overrightarrow{\mathbf{1}})(\mathbf{x},\mathbf{y}) = 1\}}{|\mathcal{S}_r(\mathbf{S}_\mathcal{X}, \mathbf{S}_\mathcal{Y}, \overrightarrow{\mathbf{1}})|} \leq 1/2 \right] > \epsilon \right]
$$

$$
= \mathop{\mathbb{P}}_{\mathbf{S} \sim \mathcal{D}_1^m} \left[ \mathop{\mathbb{P}}_{(\mathbf{x},\mathbf{y},\mathbf{1}) \sim \mathcal{D}_1} \left[ \sum_{\mathbf{S}^* \in \mathcal{S}_r(\mathbf{S})} \frac{\mathbb{1}\{\bar{\mathcal{A}}(\mathbf{S}^*)(\mathbf{x},\mathbf{y}) = 1\}}{|\mathcal{S}_r(\mathbf{S})|} \leq 1/2 \right] > \epsilon \right]
$$

$$
\leq \mathop{\mathbb{P}}_{\mathbf{S} \sim \mathcal{D}_1^m} \left[ \mathcal{L}_{\mathcal{D}_1} \left( \mathrm{Maj}_\perp(\mathcal{A}(\mathcal{S}_r(\mathbf{S}))) \right) > \epsilon \right],
$$

which concludes the proof of the lemma. ∎

### 3.2. Lower Bound

We now turn our attention to negative results which underscore the limits of majorities in learning classes of large Graph dimension yet small DS dimension. The following lower bound borrows ideas from the *first Cantor class* construction of Daniely et al. (2015) and the coupon collector lower bound of Auer and Ortner (2007). Let us briefly describe each of these lower bounds and how our approach differs from them.

The lower bound of Auer and Ortner (2007) considers a universe of size $d/\varepsilon$ and defines a binary hypotheses class containing all functions which output the 1 label on at most $d$ many points. As target hypothesis, they choose the constant all-zeroes function, and they set the marginal distribution over unlabeled datapoints to be the uniform one. It is not difficult to see that this class has VC dimension $d$. Furthermore, by a coupon collector's argument, the learner upon observing $\Theta(d \ln(1/\varepsilon)/\varepsilon)$ draws has still not seen $d$ of the elements in the universe with probability $1/2$. From this, one can define a bad ERM learner which outputs the 1 label on as many non-training points as possible. By the previous reasoning, one can conclude that this bad ERM learner in general requires $\Omega(d \ln(1/\varepsilon)/\varepsilon)$ samples in order to attain error $\leq \epsilon$, in contrast to the above majority voters $\frac{d}{\varepsilon}$. As binary classification is technically a special case of multiclass classification, in which $|\mathcal{Y}| = 2$, this result trivially gives a lower bound for the multiclass setting. However, when $|\mathcal{Y}| = 2$ the DS dimension and Graph dimension are equal to each other (and to the VC dimension), and thus the previous construction yields problems which are arbitrarily difficult to learn as $d \to \infty$. In order to describe the limits of ERM's majorities, it would be more insightful to instead design a sequence of classes $\{\mathcal{H}_d\}_{d \in \mathbf{N}}$ with Graph dimension $d$ and *constant* DS dimension, for which the performance of ERM's majorities can be lower bounded using $d$. Notably, this would yield a separation between the sample complexities of learning with one ERM, learning with ERM's majorities, and of learning with arbitrary learners.

To this end, we draw inspiration from the first Cantor class of (Daniely and Shalev-Shwartz, 2014; Daniely et al., 2015), for which each hypothesis $h_A$ is identified by a subset $A \subseteq \mathcal{X}$ and $h(x)$ equals $A$ if $x \in A$ and $*$ otherwise. Thus, we can show that $d_{DS} = 1$ and $d_G = d$, and it is still the case that there exists a bad ERM learner that needs $d \ln(1/\varepsilon)/\varepsilon$ samples to get $\varepsilon$ error when

we set the target hypothesis to be the all $*$ function. Interestingly, the fact that the labels for each hypothesis/subset are unique outside $*$ also allows us to show a lower bound for this same instance of Graph dimension $d_G$, for any majority voter which is a combination of the bad ERM learner on any partitioning of the input sample $\mathbf{S}$. This rules out the possibility that the positive results from binary classification of majority voters of any ERM-learner on smartly chosen sub-training sequences can be generalized to the multiclass setting, where learnability is characterized by the finiteness of $d_{DS}$; in the above construction $d_{DS} = 1$ and we can make the Graph dimension as large as we desire. We now give the formal statement of the result.

**Theorem 10** *For any $d \in \mathbf{N}$, $\varepsilon \leq 1/100$, and $\delta \leq 1/2$, there exists a domain $\mathcal{X}$, label set $\mathcal{Y}$, hypothesis class $\mathcal{H}$ with Graph-dimension $d_G(\mathcal{H}) = d$ and DS-dimension $d_{DS}(\mathcal{H}) = 1$, a hard distribution $\mathcal{D}$ and a bad ERM-algorithm $\mathcal{A}_{bad}$ such that for a training sequence $\mathbf{S} \sim \mathcal{D}^m$ and any number $n$ of sub-training sequences $\mathbf{S}_1, \ldots, \mathbf{S}_n$ of $\mathbf{S}$ in order to attain*

$$\mathop{\mathbb{P}}_{\mathbf{S}\sim\mathcal{D}^m} \Big[ \mathcal{L}_{\mathcal{D}} \Big( \mathrm{Maj}\big(\mathcal{A}_{bad}(\mathbf{S}_1), \ldots, \mathcal{A}_{bad}(\mathbf{S}_n)\big)\Big) \leq \varepsilon \Big] \geq 1 - \delta \,,$$

*it must be that*

$$m = \Omega\left(\frac{d + \ln(1/\delta)}{\varepsilon}\right).$$

*Moreover, attaining*

$$\mathop{\mathbb{P}}_{\mathbf{S}\sim\mathcal{D}^m} \Big[ \mathcal{L}_{\mathcal{D}}\big(\mathcal{A}_{bad}\big) \leq \varepsilon \Big] \geq 1 - \delta$$

*requires that*

$$m = \Omega\left(\frac{d\ln(1/\varepsilon) + \ln(1/\delta)}{\varepsilon}\right).$$

Before giving the proof of Theorem 10 we make a small remark. In the proof of the above, we in fact demonstrate that when $\mathrm{Maj}(\mathcal{A}_{bad}(\mathbf{S}_1), \ldots, \mathcal{A}_{bad}(\mathbf{S}_n))$ fails, it is the case that $\mathcal{A}_{bad}(\mathbf{S}_1), \ldots, \mathcal{A}_{bad}(\mathbf{S}_n)$ does not contain the correct label. That is, the lower bound shows that if we instead were to view $\mathcal{A}_{bad}(\mathbf{S}_1), \ldots, \mathcal{A}_{bad}(\mathbf{S}_n)$ as a list of candidate labels, and to only require that the correct label should be among this list, we would arrive at the same lower bound. This problem of learning a list that contains the correct label is known as *list learning* (Brukhim et al., 2022) – thus our lower bound also implies a lower bound of $\Omega((d_G + \ln(1/\delta))/m)$ for $\mathcal{A}_{bad}(\mathbf{S}_1), \ldots, \mathcal{A}_{bad}(\mathbf{S}_n)$ as a list learner, independently of the number $n$.

**Proof of Theorem 10** Consider the universe $\mathcal{X} = [\lceil d/(4\varepsilon)\rceil]$ for $\varepsilon \leq 1/(8\cdot\exp(\sqrt{(2)}+1))$, integer $d \geq 1$ and $\delta \leq 1/2$. Let $\binom{\mathcal{X}}{\leq d}$ denote all subsets of size at most $d$ from $\mathcal{X}$, i.e., $\{A | A \subseteq \mathcal{X}, |A| \leq d\}$ and consider the label space $\mathcal{Y} = \binom{\mathcal{X}}{\leq d} \cup \{*\}$. Now for $A \in \binom{\mathcal{X}}{\leq d}$ define $h_A$ to be the hypothesis which given input point $x$ outputs $A$ if $x \in A$ and $*$ else. Let $\mathcal{H}$ be the hypothesis class consisting of all such $h_A$ for $A \in \binom{\mathcal{X}}{\leq d}$, i.e., $\mathcal{H} = \{h_A\}_{A\in\binom{\mathcal{X}}{\leq d}}$. Furthermore, let $f^*$ be the all $*$'s ($f^* = *$) and notice that this hypothesis is in $\mathcal{H}$ as the empty set.

We notice that the DS-dimension of this hypothesis class is 1. To see this assume that two points $x_1$ and $x_2$ are DS-shattered by $\mathcal{H}$, then it is the case that there exists some non-empty $H' \subseteq \mathcal{H}$ such

that for every $h \in H'$ and $i \in 1,2$, there exists $g \in H'$ such that $h(x_i) \neq g(x_i)$ and $h(x_j) = g(x_j)$ for $j \neq i$. Thus, for $h \in H'$ and $i = 1$, there exists $g \in H'$ such that $h(x_1) \neq g(x_1)$ and $h(x_2) = g(x_2)$. Hence, the first condition implies that either $h$ or $g$ is not $*$ on $x_1$; assume without loss of generality it is $h$. Now for $h$, there also exists some $g' \in H'$ such that $h(x_1) = g'(x_1)$ and $h(x_2) \neq g'(x_2)$ but this cannot be the case as we had that $g'(x_1) = h(x_1) \neq *$, which since the label of a function in $\mathcal{H}$ uniquely characterize the function implies that $g' = h$, thus contradicts $h(x_2) \neq g'(x_2)$. We thus conclude that the DS-dimension must be strictly less than 2. It is also at least 1 since any point $x_1 \in \mathcal{X}$ can be DS-shattered by $H' = \{f^*, f_{x_1}\}$.

We now notice that the Graph-dimension of this class is $d$. To see this we first notice that any $d$ distinct points $x_1, \ldots, x_d \in \mathcal{X}$ can be Graph-shattered, with $f^*$ as witness, since for any $b \in \{0,1\}^d$, the function $f_A$, with $A = \{x \in \mathcal{X} | x = x_i \text{ and } b_i = 0\}$, will for $b_i = 0$ output $A = f(x_i) \neq f^*(x_i)$ and for $b_i = 1$ output $* = f(x_i) = f^*(x_i)$, thus $f^*$ can be Graph shattered on any $d$ distinct points. Further assume there exists $f$ and points $x_1, \ldots, x_{d+1}$ where $f$ is Graph shattered. Then there exists a function $h \in \mathcal{H}$ which is equal to $f$ on $x_1, \ldots, x_{d+1}$. We notice that this function $h$ must necessarily be all $*$'s on $x_1 \ldots, x_{d+1}$. To see this, we first notice that since any $h \in \mathcal{H}$ can have at most $d$ non $*$ values there must exist $x_i \in \{x_1, \ldots, x_{d+1}\}$ such that $h(x_i) = *$. Further if $h$ is $y \in \mathcal{Y} \backslash \{*\}$ for some $x_j \in \{x_1, \ldots, x_{d+1}\}$, this function can not be Graph shattered, since any $b \in \{0,1\}^{d+1}$ with $b_i = 0$ and $b_j = 1$ can not be realized by a function in $\mathcal{H}$, since such a function because $b_i = 0$, should be $\neq *$ for $x_i$, so must be some $y' \in \mathcal{Y}$, and by $b_j = 1$ should be $= y$, but since the labels of the functions are unique for the function in $\mathcal{H}$ such a function in $\mathcal{H}$ must have $y' = y$ and also be equal to $h$, however $h$ is not equal to $y$ on $x_i$ it is $*$ so we reach a contradiction. Thus $h$ has to be all $*$'s on $x_1, \ldots, x_{d+1}$ but then $b = (0, \ldots, 0)$ can not be realized since no function can output more than $d$ non $*$-values.

We now consider the distribution $\mathcal{D}_U$ which assigns uniform mass to each element in $\mathcal{X}$, i.e., $\mathcal{D}_U(x) = 1/|\mathcal{X}|$ for $x \in \mathcal{X}$. We will further use $\mathcal{D}_{U,f^*}$ as the distribution over $\mathcal{X} \times \mathcal{Y}$ that with $\mathcal{D}_{U,f^*}[x, f^*(x)] = 1/|\mathcal{X}|$ for $x \in \mathcal{X}$. For a training sequence $S$ we define the points of $\mathcal{X}$ not in $S$ as $\mathcal{X}_{unseen,S} = \{x \in \mathcal{X} : x \neq S_j, \forall j \in [|S|]\}$. Now the bad ERM $\text{ERM}_{bad}$ does as follows upon being given a training sequence $S$: If $|\mathcal{X}_{unseen,S}|$ is larger than or equal to $d$ it takes the $d$ smallest integers of $\mathcal{X}_{unseen,S}$ and denotes these points $A_S$ and outputs $f_{A_S}$. If $|\mathcal{X}_{unseen,S}|$ is less than $d$ it outputs $f_{|\mathcal{X}_{unseen,S}|}$. That is, formally $A_S = \cup_{i=1}^{l}([i] \cap \mathcal{X}_{unseen,S})$ for the smallest $l \leq |\mathcal{X}|$ such that $\cup_{i=1}^{l}([i] \cap \mathcal{X}_{unseen,S}) \leq d$ and if $l+1 \in \mathcal{X}$ then $\cup_{i=1}^{l+1}([i] \cap \mathcal{X}_{unseen,S}) > d$. We notice that $\text{ERM}_{bad}$ is indeed an ERM-learner as it is consistent with $f^*$ on $S$ and it is proper as it is in $\mathcal{H}$ by the above described $A$ always having size less than or equal to $d$.

We first consider the lower bound for the majority voter, where the $\text{ERM}_{bad}$ is run on $n$ different sub-training sequences $\mathbf{S}^i$ of $\mathbf{S} \sim \mathcal{D}_{U,f^*}^m$, i.e., each training example in $\mathbf{S}^i$ is in $\mathbf{S}$, however, it may occur in $\mathbf{S}^i$ a different number of times. We consider the classifier $\text{Maj}(\text{ERM}_{bad}(\mathbf{S}^1), \ldots, \text{ERM}_{bad}(\mathbf{S}^n))$. We consider two cases $m \leq d/(16\varepsilon)$ and $m \leq \ln(1/\delta)/(8\varepsilon)$ and show that in both cases with probability at least $\delta$ the majority voter has error $\varepsilon$, whereby concluding that to have error less than $\varepsilon$ it must be the case that $m \geq \max(d/(16\varepsilon), \ln(1/\delta)/(8\varepsilon))$. We start with the former case.

If $m \leq d/(16\varepsilon)$ we consider the first $d$ elements of the universe $\mathcal{X}$, i.e., $[d]$. We Notice that the elements $[d]$ has $d/\lceil d/4\varepsilon \rceil \leq 4\varepsilon$ mass. Thus if we define $\mathbf{X} = \sum_{i=1}^{m} \mathbb{1}\{\mathbf{S}_i \in [d]\}$ the expected number of examples in the training sequence $\mathbf{S}$ that lands in $[d]$ is $\mathbb{E}[\mathbf{X}] \leq 4\varepsilon m \leq 4d/16$ since we assumed that $m \leq d/(16\varepsilon)$. Thus, by Markov's inequality we have that $\mathbb{P}[X \geq d/2] \leq 2\mathbb{E}[X]/d \leq 8/16 = 1/2$.

Thus, with probability at least $1/2$ there are strictly less than $d/2$ training examples of $\mathbf{S}$ in $[d]$, that is $|\mathcal{X}_{unseen,\mathbf{S}} \cap [d]| > d/2$. Now, consider any realization $S$ of $\mathbf{S}$ such that $|\mathcal{X}_{unseen,S} \cap [d]| > d/2$. Thus, for any such event any $\mathrm{ERM}_{bad}(S^i)$ will not output $*$ on points $x \in \mathcal{X}_{unseen,S} \cap [d]$, since $S^i$ is a sub-training sequence of $S$ so $\mathcal{X}_{unseen,S} \cap [d] \subseteq \mathcal{X}_{unseen,S^i} \cap [d]$ i.e. $d/2 \leq |\mathcal{X}_{unseen,S^i} \cap [d]| \leq d$, and since $\mathrm{ERM}_{bad}(\mathbf{S}^i)$ outputs $f_{A_{\mathbf{S}^i}}$, where $A_{\mathbf{S}^i} = \cup_{i=1}^{l}([i] \cap \mathcal{X}_{unseen,\mathbf{S}^i})$ for the smallest $l \leq |\mathcal{X}|$ such that $\cup_{i=1}^{l}([i] \cap \mathcal{X}_{unseen,\mathbf{S}^i}) \leq d$ and if $l+1 \in \mathcal{X}$ then $\cup_{i=1}^{l}([i] \cap \mathcal{X}_{unseen,\mathbf{S}^i}) > d$, it must be the case that $l \geq d$ and that $\mathcal{X}_{unseen,S} \cap [d] \subseteq \mathcal{X}_{unseen,S^i} \cap [d] \subseteq A_{\mathbf{S}^i}$.

Thus, for any $x \in \mathcal{X}_{unseen,S} \cap [d]$ and any $i = 1, \ldots, n$ we have $\mathrm{ERM}_{bad}(S^i)(x) \neq *$ so $\mathrm{Maj}(\mathrm{ERM}_{bad}(S^1), \ldots, \mathrm{ERM}_{bad}(\mathbf{S}^n))$ fails for all $x \in \mathcal{X}_{unseen,S} \cap [d]$ as it never sees a $*$ for $x \in \mathcal{X}_{unseen,S} \cap [d]$ which has size strictly more than $d/2$. Hence, we conclude that

$$\mathcal{L}_{\mathcal{D}_{U,f^*}}(\mathrm{Maj}(\mathrm{ERM}_{bad}(S^1), \ldots, \mathrm{ERM}_{bad}(S^n))) > (d/2)/\lceil d/(4\varepsilon)\rceil \geq (d/2)/(2d/(4\varepsilon)) \geq \varepsilon$$

for any realization $S$ of $\mathbf{S}$ such that $|\mathcal{X}_{unseen,\mathbf{S}} \cap [d]| > d/2$, and since this happens with probability $1/2 \geq \delta$, we are done for the case $m \leq d/(16\varepsilon)$.

Now, for the case that $m \leq \ln(1/\delta)/(8\varepsilon)$ we again have that $[d]$ has $d/\lceil d/(4\varepsilon)\rceil \leq 4\varepsilon$ mass, thus

$$\mathbb{P}_{\mathbf{S}\sim\mathcal{D}_{U,f^*}^m}[|\mathbf{S} \cap [d]| = 0] = \mathbb{P}_{\mathbf{S}\sim\mathcal{D}_{U,f^*}^m}[\forall i \in [m]\ \mathbf{S}_i \notin [d]] \geq (1-4\varepsilon)^m = \exp(\ln(1-4\varepsilon)m),$$

and since $\ln(1-x) \geq -2x$ for $x \leq 1/2$, and $m \leq \ln(1/\delta)/(8\varepsilon)$ and $\varepsilon \leq 1/8$ we conclude that $\mathbb{P}_{\mathbf{S}\sim\mathcal{D}_{U,f^*}^m}[|\mathbf{S} \cap [d]| = 0] \geq \exp(-8\varepsilon m) \geq \delta$. But for an outcome $S$ of $\mathbf{S}$ such that $|\mathbf{S} \cap [d]| = 0$ we have that $\mathcal{X}_{unseen,S} \cap [d] = [d]$ and thus $[d] = \mathcal{X}_{unseen,S} \cap [d] \subseteq \mathcal{X}_{unseen,S^i} \cap [d]$ for any $i \in [n]$ which implies that $\mathrm{ERM}_{bad}(S^i) = f_{[d]}$ for any $i$ and for $x \in [d]$, we thus have $\mathrm{Maj}(\mathrm{ERM}_{bad}(S^1), \ldots, \mathrm{ERM}_{bad}(S^n))(x) = [d] \neq *$ and as $d/\lceil d/(4\varepsilon)\rceil \geq d/(2d/(4\varepsilon)) \geq 2\varepsilon$ we conclude that $\mathcal{L}_{\mathcal{D}_{U,f^*}}(\mathrm{Maj}(\mathrm{ERM}_{bad}(S^1), \ldots, \mathrm{ERM}_{bad}(S^n))) \geq 2\varepsilon$ for any realization of $S$ of $\mathbf{S}$ such that $|S \cap [d]| = 0$, and since this happens with at least $\delta$ probability, we are done and conclude the claim for the sample complexity of any majority voter of $\mathrm{ERM}_{bad}$ on sub-training sequences.

For the lower bound of $\mathrm{ERM}_{bad}(\mathbf{S})$, we notice that the above also gives the bound of $m \geq \ln(1/\delta)/(8\varepsilon)$ as $\mathrm{ERM}(\mathbf{S}) = \mathrm{Maj}(\mathrm{ERM}(\mathbf{S}))$.

Now, for the case that $m \leq d\ln(1/(8\exp(\sqrt{2})\varepsilon))/(4\varepsilon)$. We see the above as a coupon collector's problem. Define $N_i$ to be the number of draws one should make from $\mathcal{D}_{U,f^*}$ before seeing a new example in $\mathcal{X}$ upon already having seen $i-1$ different examples of $\mathcal{X}$ for $i \geq 1$. We then see that the $N_i$'s are geometrically distributed with success probability $(|\mathcal{X}|-i+1)/|\mathcal{X}|$. Now, consider the random variable $\mathbf{N} = \sum_{i=1}^{|\mathcal{X}|-d} N_i$, which is counting the number of draws from $\mathcal{D}_{U,f^*}$ we need to make before seeing $|\mathcal{X}| - d$ different points in $\mathcal{X}$. Now, using that $\{|\mathcal{X}|-i+1\}_{i=1}^{|\mathcal{X}|-d}$ is equal to $\{i\}_{i=d+1}^{|\mathcal{X}|}$ in reverse order and for a decreasing function $f$ we have that $\sum_{i=a}^{b} f(i) \geq \int_{a}^{b+1} f(i)\,di$ we get that the expected value of $\mathbf{N}$ is

$$\mathbb{E}[\mathbf{N}] = \sum_{i=1}^{|\mathcal{X}|-d} \frac{|\mathcal{X}|}{|\mathcal{X}|-i+1} = |\mathcal{X}| \sum_{i=d+1}^{|\mathcal{X}|} \frac{1}{i} \geq |\mathcal{X}| \int_{d+1}^{|\mathcal{X}|+1} \frac{1}{i}\,di = |\mathcal{X}| \ln\left(\frac{|\mathcal{X}|+1}{d+1}\right) \qquad (1)$$

and since $N_i$ and $N_j$ for $j > i$ is such that $\mathbb{E}_{N_i,N_j}\left[N_i N_j\right] = \mathbb{E}_{\mathbf{N}_i}\left[N_i \mathbb{E}_{N_j}\left[N_j | \mathbf{N}_i\right]\right] = \mathbb{E}_{\mathbf{N}_i}\left[N_i\right] \mathbb{E}_{N_j}\left[N_j\right]$ so the $\mathrm{Cov}(\mathbf{N}_i, \mathbf{N}_j) = 0$, we get that

$$\mathrm{Var}(\mathbf{N}) = \sum_{i=1}^{|\mathcal{X}|-d} \frac{|\mathcal{X}|\,(i-1)}{(|\mathcal{X}| - i + 1)^2} = -|\mathcal{X}| \sum_{i=1}^{|\mathcal{X}|-d} \frac{(|\mathcal{X}| - i + 1 - |\mathcal{X}|)}{(|\mathcal{X}| - i + 1)^2} = |\mathcal{X}| \sum_{i=d+1}^{|\mathcal{X}|} \frac{|\mathcal{X}| - i}{i^2} \quad (2)$$

$$\leq |\mathcal{X}|^2 \int_{i=d}^{|\mathcal{X}|} \frac{1}{i^2}\, \mathrm{d}i \leq \frac{|\mathcal{X}|^2}{d}. \quad (3)$$

Thus, we have by Markov's inequality that

$$\mathbb{P}\left[|\mathbf{N} - \mathbb{E}\left[\mathbf{N}\right]| \geq \sqrt{2\mathrm{Var}(\mathbf{N})}\right] \leq \frac{1}{2},$$

i.e., with probability at least $1/2 \geq \delta$ we have that $\mathbf{N} \geq \mathbb{E}\left[\mathbf{N}\right] - \sqrt{2\mathrm{Var}(\mathbf{N})}$. Using the bounds in Eq. (1) and Eq. (2), along with $|\mathcal{X}| = \lceil d/(4\varepsilon)\rceil$, then implies that

$$\mathbb{E}\left[\mathbf{N}\right] - \sqrt{2\mathrm{Var}(\mathbf{N})} \geq |\mathcal{X}|\left(\ln\left(\frac{|\mathcal{X}|+1}{d+1}\right) - \sqrt{\frac{2}{d}}\right) \geq \frac{d}{4\varepsilon}\left(\ln\left(\frac{1}{8\varepsilon}\right) - \sqrt{2}\right)$$

$$= \frac{d}{4\varepsilon} \ln\left(\frac{1}{8\exp\left(\sqrt{2}\right)\varepsilon}\right).$$

Thus, we conclude that $\mathbf{N} > d\ln\left(1/8\exp\left(\sqrt{2}\right)\varepsilon\right)/(4\varepsilon)$ with probability at least $1/2 \geq \delta$. That is, if $m \leq d\ln\left(1/8\exp\left(\sqrt{2}\right)\varepsilon\right)/(4\varepsilon)$ then with probability at least $1/2$ we have that $\mathcal{X}_{unseen,S}$ is at least of size $d$ thus $|A_S|$ is at least $d$, and thus $\mathrm{ERM}_{bad}(S) = f_{A_S}$ will fail on all points $x \in A_S$, so have $\mathcal{L}_{\mathcal{D}_{U,f^*}}(\mathrm{ERM}_{bad}(S)) \geq d/|\mathcal{X}| \geq d/2d/(4\varepsilon) \geq 2\varepsilon$ which concludes the case $m \leq d\ln\left(1/8\exp\left(\sqrt{2}\right)\varepsilon\right)/(4\varepsilon)$ and the claim in the lemma. ∎

## 4. Limitations of Aggregating Proper Learners

The results of Section 3 underscore the strength of majorities of ERM's in learning classes of finite graph dimension — along with their weakness in learning classes of infinite graph dimension which are nevertheless learnable (i.e., which have finite DS dimension). It is natural to ask, then, whether a more sophisticated aggregation of proper learners might have the capacity to learn all classes of finite DS dimension. Perhaps by using more elaborate proper learners than ERM, or a more subtle aggregation strategy than majority voting, one can learn any DS class with an aggregation of proper learners?

We now demonstrate this is not the case: there exist learnable classification problems that cannot be learned by *any* combination of a finite number of proper learners. Informally, we demonstrate that proper learning cannot serve as a foundation for multiclass learning in its full generality, at least not in the most direct sense. It may be that classification problems, in their most difficult forms, require techniques that are fundamentally more complex than those offered by the perspective of proper learning.

Formalizing these notions relies upon the design of *properness numbers*, which measure the complexity of a function $f \colon \mathcal{X} \to \mathcal{Y}$ relative to an underlying hypothesis class $\mathcal{H}$.

**Definition 11** *Let $\mathcal{H} \subseteq \mathcal{Y}^{\mathcal{X}}$ be a hypothesis class and $f \colon \mathcal{X} \to \mathcal{Y}$ a function. The **properness number** of $f$ with respect to $\mathcal{H}$, denoted $\mathsf{prop}_{\mathcal{H}}(f)$, equals the size of the smallest finite set $H \subseteq \mathcal{H}$ such that*

$$f(x) \in \{h(x) : h \in H\} \quad \forall x \in \mathcal{X}.$$

*In the event that no such finite $H \subseteq \mathcal{H}$ exists, we set $\mathsf{prop}_{\mathcal{H}}(f) = \infty$.*

Geometrically, $\mathsf{prop}_{\mathcal{H}}(f)$ measures the number of distinct functions in $\mathcal{H}$ which must be "interwoven" or "pieced together" in order to assemble $f$. Note, however, that $\mathsf{prop}_{\mathcal{H}}(f)$ does not track the number of different *regions* on which $f$ alternates between different functions in $\mathcal{H}$. It may well be that $f$ is a piecewise combination of a relatively small number of functions in $\mathcal{H}$, yet alternates between them in a highly complex way. (Ultimately, this reflects our agnostic perspective on the *aggregation strategy* for combining several proper learners into one improper learner, which is permitted to be arbitrarily complex.) This point is emphasized by the following example.

**Example 1** *Let $\mathcal{X} = \mathbb{N}$, $\mathcal{Y} = \{0,1\}$, and define $\mathcal{H}$ to contain only the two constant functions. Then any function $f : \mathcal{X} \to \mathcal{Y}$ has $\mathsf{prop}_{\mathcal{H}}(f) \leq 2$, as the two functions in $\mathcal{H}$ span all labels in $\mathcal{Y}$ at any input $x \in \mathcal{X}$.*

Note too, of course, that a function $f : \mathcal{X} \to \mathcal{Y}$ has $\mathsf{prop}_{\mathcal{H}}(f) = 1$ if and only if $f \in \mathcal{H}$. We now extend the notion of properness numbers to learners, in the natural way.

**Definition 12** *Let $\mathcal{H} \subseteq \mathcal{Y}^{\mathcal{X}}$ be a hypothesis class and let $A$ be a learner for $\mathcal{H}$. Use $\mathcal{S}_{\mathcal{H}}$ to denote the collection of all $\mathcal{H}$-realizable samples, i.e., $\mathcal{S}_{\mathcal{H}} = \{S \in (\mathcal{X} \times \mathcal{Y})^* : \min_{h \in \mathcal{H}} L_S(h) = 0\}$. Then*

$$\mathsf{prop}_{\mathcal{H}}(A) = \sup_{S \in \mathcal{S}_{\mathcal{H}}} \mathsf{prop}_{\mathcal{H}}(A(S)).$$

We now demonstrate the central result of the section: there exist learnable hypothesis classes $\mathcal{H}$ for which *any* learner $A$ for $\mathcal{H}$ must have $\mathsf{prop}_{\mathcal{H}}(A) = \infty$. That is, $A$ must emit functions which cannot be realized as aggregations of a bounded number of hypotheses in $\mathcal{H}$. Consequently, they cannot be realized as aggregations of outputs from a bounded number of proper learners for $\mathcal{H}$.

**Theorem 13** *Let $\mathcal{X}$ and $\mathcal{Y}$ be infinite sets. Then there exists an $\mathcal{H} \subseteq \mathcal{Y}^{\mathcal{X}}$ with DS dimension 1 and the property that any PAC learner $\mathcal{A}$ for $\mathcal{H}$ must have $\mathsf{prop}_{\mathcal{H}}(\mathcal{A}) = \infty$. In fact, learning $\mathcal{H}$ to expected error $< \frac{1}{2}$ requires learners with infinite properness number.*

**Proof** It suffices to prove the claim for a choice of countable $\mathcal{X}$ and $\mathcal{Y}$. First, let $\{\mathcal{X}_d\}_{d \in \mathbb{N}}$ be a family of disjoint sets with $|\mathcal{X}_d| = d$, and for each such $d$ let $\mathcal{Y}_d = \{\star\} \cup 2^{\mathcal{X}_d}$, where $2^{\mathcal{X}_d}$ denotes the power set of $\mathcal{X}_d$. Then, for an $A \subseteq \mathcal{X}_d$, define $h_A \colon \mathcal{X}_d \to \mathcal{Y}_d$ by

$$h_A(x) = \begin{cases} A & x \in A, \\ \star & x \notin A. \end{cases}$$

Further, define $\mathcal{H}_d \subseteq \mathcal{Y}_d^{\mathcal{X}_d}$ as $\mathcal{H}_d = \{h_A : A \subseteq \mathcal{X}_d, |A| = d - \sqrt{d}\}$. Now set

$$\mathcal{X} = \bigcup_{d \in \mathbb{N}} \mathcal{X}_d, \qquad \mathcal{Y} = \{\$\} \cup \left( \bigcup_{d \in \mathbb{N}} \mathcal{Y}_d \right).$$

As $\mathcal{X}$ and $\mathcal{Y}$ are countable unions of finite sets, they are countable. Let $\mathcal{H} = \bigcup_{d \in \mathbb{N}} \mathcal{H}_d$, where each $h \in \mathcal{H}_d$ is extended to a total function $\mathcal{X} \to \mathcal{Y}$ by outputting the label \$ on all $x \notin \mathcal{X}_d$.

We first demonstrate that $\mathcal{H}$ has a DS dimension of 1. Begin by setting $S = \{x_1, x_2\} \subseteq \mathcal{X}$, and suppose that $S$ is DS-shattered by $\mathcal{F} \subseteq \mathcal{H}$. Let $f \in \mathcal{F}|_S$ be a behavior on $S$. Then it must be that $f(x_1), f(x_2) \in \{\$, \star\}$. Otherwise, we have without loss of generality that $f(x_1) = A$, and thus $f$ cannot have a 2-neighbor in $\mathcal{F}|_S$, as any hypothesis in $\mathcal{H}$ which emits the label $A$ must be $h_A$. (I.e, any $g \in \mathcal{F}|_S$ with $g(x_1) = f(x_1) = A$ must be $g = f$.) So we indeed have that $f \in \{\$, \star\}^2$ and thus $\mathcal{F}|_S \subseteq \{\$, \star\}^2$. Then in order for $\mathcal{F}$ to DS-shatter $S$, it must be that $\mathcal{F}|_S = \{\$, \star\}^2$. For the all-$\star$ behavior to be expressible, it must be that $S \subseteq \mathcal{X}_d$ for some $d$. (In particular, any hypothesis in $\mathcal{H}$ emits the $\star$ label on only a single $\mathcal{X}_d \subseteq \mathcal{X}$.) In this case, however, any behavior in $\{\$, \star\}^2$ other than the all-$\star$ behavior or the all-\$ behavior cannot be expressed by $\mathcal{F}|_S$, as any hypothesis in $\mathcal{H}$ either emits the \$ label on all points in $\mathcal{X}_d$ or on none of its points. Thus $d_{DS}(\mathcal{H}) = 1$.

Now let $\mathcal{A}$ be a learner with $\mathsf{prop}_{\mathcal{H}}(\mathcal{A}) = p < \infty$. Select $d$ so that $d = \omega(p\sqrt{d})$, i.e., $d = \omega(p^2)$. For a hypothesis $h_A \in \mathcal{H}_d \subseteq \mathcal{H}$, let $D_{h_A}$ denote the distribution which draws a point uniformly at random from $A^C = \mathcal{X}_d \setminus A$ and labels it using $h_A$ (i.e., labels all points in its support with the $\star$ label). Now consider the process by which an $h_A \in \mathcal{H}_d$ is chosen uniformly at random and $D_{h_A}$ is chosen to be the true distribution. In order to incur error $\leq \epsilon$, $\mathcal{A}$ must emit a function $f$ outputting the value $\star$ on all but $\epsilon \cdot \sqrt{d}$ many points in $A^C$.

As $\mathsf{prop}_{\mathcal{H}}(\mathcal{A}) \leq p$, then $\mathsf{prop}_{\mathcal{H}}(f) \leq p$ and thus $f$ can output at most $p \cdot \sqrt{d} = o(d)$ values of $\star$ on $\mathcal{X}_d$. Furthermore, by taking $d$ to be arbitrarily large, we can assume that $|S| \in o(\sqrt{d})$. Together, we have that $\mathcal{A}$ must identify a set of size $o(d)$ points containing a $(1 - \epsilon)$-fraction of the points in $A^C$, given only knowledge of a $o(1)$-fraction of points in $A^C$. As the posterior distribution of $A^C$ is uniform over all points in $\mathcal{X}_d$, by our uniformly random choice of $D_{h_A}$, the learner $\mathcal{A}$ can do no better than to predict $\star$ on a random choice of $p \cdot \sqrt{d}$ elements in $\mathcal{X}_d$, i.e., a $o(1)$-fraction of $\mathcal{X}_d$. As such, any learner $\mathcal{A}$ attains expected error $1 - o(1)$ over the uniformly random choice of $h_A \in \mathcal{H}_d$. An invocation of the probabilistic method reveals that there exists a particular "bad" choice of $h_A$ such that $\mathcal{A}$ incurs $1 - o(1)$ error on $D_{h_A}$. This concludes the argument. ∎

**Remark 14** *Let us briefly summarize the modifications required of* Daniely and Shalev-Shwartz *(2014)'s first Cantor class for the proof of Theorem 13. First, we extend each function $h_A \in \mathcal{Y}_d^{\mathcal{X}_d}$ to a total function $\mathcal{X} \to \mathcal{Y}$ using a dedicated "\$" label, rather than using the "$\star$" label as in* Daniely and Shalev-Shwartz *(2014). Otherwise, a learner can attain the constant $\star$ behavior on nearly all of $\mathcal{X} = \bigcup_{\mathbb{N}} \mathcal{X}_d$, as each hypothesis in $\mathcal{H}$ emits the $\star$ label on all subdomains $\mathcal{X}_d$ except a single one. Strictly speaking, this permits the first Cantor class to be learned with a properness number of 1 (i.e., using a proper learner). Second, the first Cantor class defines each $h_A$ such that $|A| = |\mathcal{X}_d|/2 = d/2$. This allows for a learner to attain the constant $\star$ behavior on a subdomain $\mathcal{X}_d$ by aggregating only two classifiers, namely $h_A$ and $h_{A^C}$. We instead require that the cardinality of $A^C$ be sublinear in $d$, which in turn guarantees that expressing the constant $\star$ behavior on any domain $\mathcal{X}_d$ requires aggregating $\omega_d(1)$ hypotheses.*

We conclude the section with two brief remarks on properness numbers. First, there is a connection between properness numbers and the classic principle of *boosting*, which — roughly speaking — improves the performance of one or several *weak learners* by aggregating their outputs over various subsamples of the training set (Schapire and Freund, 2012). In the event that one's weak learners

are proper (or have finite properness numbers), and if they need only be aggregated a bounded number of times as $|S|, |\mathcal{Y}| \to \infty$, then boosting may serve as an avenue towards learning with finite properness numbers. Notably, the theory of multiclass boosting is somewhat less developed than in the binary case, with both negative results describing its limits and promising work exhibiting boosting rules whose complexities do not depend upon $|\mathcal{Y}|$ (see, e.g., Saberian and Vasconcelos (2011); Brukhim et al. (2021, 2023, 2024)).

Our second remark on properness numbers, related to the first: it may be natural to consider the properness number of a learner $\mathcal{A}$ as a *sequence* indexed by $|S|$. That is, a learner $\mathcal{A}$ may have infinite properness number (as defined in Definition 12) yet for each $n \in \mathbb{N}$ output functions with bounded properness numbers across all $S \in (\mathcal{X} \times \mathcal{Y})^n$. In this setting, one can ask how the properness numbers of $\mathcal{A}$ scale with $|S|$, or how they scale with error parameters $\epsilon$ and $\delta$. Notably, Theorem 13 is robust to this form of relaxation, as it demonstrates a problem which requires infinite properness number merely to attain *constant* error. Nevertheless, considering properness numbers which scale with $|S|$ or with $\epsilon$ may raise interesting questions concerning the *properness complexity* of learning (whenever learning with finite properness numbers is possible).

## 5. Conclusion

In this work we have studied natural extensions of optimal learners for binary classification to the multiclass setting. Our results show that these learners have better sample complexity than simple ERM learners for certain classes with finite Graph dimension. Moreover, we have shown that there are learnable multiclass problems for which there does not exist a learner that can be expressed as any combination of finitely many proper learners. One interesting open question is to understand whether there exists a combinatorial dimension whose finiteness characterizes the existence of optimal *proper* learners for the underlying multiclass problem, in the same way that the Dual-Helly number characterizes optimality of proper learners for binary classification (Bousquet et al., 2020).

## Acknowledgments

The authors thanks Shaddin Dughmi for several insightful conversations on the topic of this research. Julian Asilis was supported by the Simons Foundation and by the National Science Foundation Graduate Research Fellowship Program under Grant No. DGE-1842487. Mikael Møller Høgsgaard was supported by Independent Research Fund Denmark (DFF) Sapere Aude Research Leader Grant No. 9064-00068B. Grigoris Velegkas was supported by the AI Institute for Learning-Enabled Optimization at Scale (TILOS). Any opinions, findings, conclusions, or recommendations expressed in this material are those of the authors and do not necessarily reflect the views of any of the sponsors such as the NSF.

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
