# OpenReview forum: "Understanding Aggregations of Proper Learners in Multiclass Classification"
_algorithmiclearningtheory.org/ALT/2025/Conference — ALT 2025_

### Official Review · Reviewer_W4YY · 2024-11-10
**This paper studies aggregation of ERMs and more generally proper learners in the setting of multiclass learning and proves new negative results.**

**Rating:** 7
**Confidence:** 4

**Review:**

Summary: This paper studies multi-class learning in the classic PAC setup which generalizes the more familiar binary classification setting. The authors investigate the role of proper learning strategies and their aggregations. Their main motivation for doing so is because it has long been known that very reasonable proper learning strategies (specifically \emph{any} empirical risk minimization (ERM) algorithm) are nearly optimal up to a logarithmic factor, and more recently it has been demonstrated that "simple" combinations of these strategies via majority vote classifiers learned on subsets of the full training sample can remove this log factor, and are hence optimal. The authors state that there are three main contributions in their paper:

1) The authors show that applying the aforementioned optimal majority voting on top of vanilla (multiclass) ERM also provides a way to achieve tighter and ``log free'' sample complexity bounds based on the Graph dimension, which is one of many generalizations of the VC dimension. This result is analogous to the binary classification setting as the Graph dimension is the notion that naturally allows one to obtain ERM learning guarantee (i.e. uniform convergence bounds) for multiclass learning.

2) The authors show that for any choice of graph dimension d, there is is a hypothesis class $\mathcal{H}_d$ with DS dimension 1 (hence learnable by a recent characterization of multiclass learnability) target function $h^\star$ and hard realizable distribution such that: (i) There is a bad choice of ERM algorithm that requires at least $\Omega(d\log(1/\epsilon)/\epsilon + \log(1/\delta)/\epsilon)$ samples to learn, mirroring the lower bound for ERM in the binary setting. Their construction is based on combining the usual hard class for multiclass learning called the "Cantor class" with a coupon collector argument. Previously, the only lower bound for such bad ERMs  did not have this log factor from below. (ii) Show that any majority voter based on this bad ERM algorithm that aggregates any arbitrary number of functions learned using this ERM algorithm on subsamples of the data must suffer error $\Omega(d/\epsilon + \log(1/\delta)/\epsilon)$. I believe this matches a comparable lower bound by Daniely and Shalev-Shwartz 2015 that holds for proper learners, though I don't believe the authors make this comparison, at least with reference to the graph dimension?

3) The authors study a more abstract notion of aggregating proper learners. Importantly, their notion of aggregation cannot always be expressed as majority votes. Their main result in this setting is showing that there are learnable function classes for which any learner must have infinite ``properness number" which is the quantitative measure of how improper a learning algorithm is that the authors introduce.

In summary, their results improve our understanding of the failure modes of using proper learner as a fundamental building blocks in learning strategies employed in the multiclass setting.

The paper is very clear and the proofs are very easy to follow. It is good that the gaps in the sample complexity upper/lower bounds for the \emph{worst case} ERM have been resolved. In particular, it is very nice to see that the delicate combination of the classic coupon collector bound and the typical ``hard'' instance for ERM in the multiclass setting works out. Also, the authors more general result regarding more abstract aggregations of proper learners being bad learning strategies is interesting.

Regarding the upper bound detailed in contribution 1) above, it seems that up to some very minor technicalities, the proof is almost an immediate consequence of the results in the binary case. This is fine for the sake of comparing to the matching lower bound for majorities in contribution 2), but I am bit confused why this is phrased as a contribution on the level of the other main results?

The authors says "It is natural to ask whether taking majorities of ERMs can help us learn classes that have infinite Graph dimension". As the authors are aware, it is well known by now that for the worst case ERM, the sample complexity scales like $\Omega(d/\epsilon)$ (which the authors improved upon in contribution 2). Given that we we should view this majority voting as machinery built in order to shave off the log factor that stems from uniform convergence, I don't see why this would be a natural question to ask.

Finally, given the strong similarity to existing lower bound construction (e.g. Daniely et al. 2015 in your references) to both lower bounds, it would be nice to also compare and contrast the lower bound construction in contribution 3 to that of existing work that focused on proving lower bounds for proper learners (as was done for contribution 2).

Some very minor comments on presentation:

in leamma 6, you may want to not use $y$ inside the probability $\mathbf{P}_{S \sim \mathcal{D}}[L_D > y]$ since it is confusing to also use it for labels. Also in the proof, whenever you use the upper bound when translating from majority vote failing to the average classifier failing, you use an equality rather than an inequality.
In the proof of theorem 13 you use some notation for projection $\mathcal{F}|_S$ that I dont think was introduced?

**Paper Award:**

No

---

> ### Author Response · Authors · 2024-11-22
>
> We thank the reviewer for their detailed feedback and careful reading of the paper.
>
> Regarding the comment on majorities of ERMs for learning classes of infinite Graph dimension, we agree that this presentation and motivation can be improved. We will reword the relevant paragraph to emphasize the sample complexity of ERM’s majorities with respect to classes of finite graph dimension, rather than the case in which the graph dimension is infinite. Thank you for this comment.
>
> Regarding the relation to the lower bound construction of Daniely et al. 2015, we agree that this merits additional discussion. We will add a remark to Section 4 describing how Theorem 13 builds upon the first Cantor class (and why the alterations made to the first Cantor class for the sake of proving Theorem 13 are necessary). In particular, the first Cantor class can technically be learned with a properness number of 1, as each hypothesis $h_A$ is defined to output the “\*” label outside all domains $\mathcal{X}_i$, save for the single domain $\mathcal{X}_d$ for which $A \subseteq \mathcal{X}_d$. This permits a learner which observes only the “\*” label in its training set to output the hypothesis $h_A$ for $A \subseteq \mathcal{X}_d$ and $d \in \mathbb{N}$ sufficiently large. For this reason, we modify the class such that each function is extended using the dedicated “\\$” label. Secondly, the first Cantor class defines each $h_A$ such that $|A| = |\mathcal{X}_d| / 2 = d / 2$ .  This allows a learner to attain the all “\*” behavior on a subdomain $\mathcal{X}_d$ by only aggregating two classifiers : one that corresponds to $A$ and one to its complement $A^C$. By requiring that the cardinality of $A$ be sublinear in that of $\mathcal{X}_d$, we resolve this second issue.
>
>
> We appreciate the minor comments on presentations, all of which will be adopted in the next version of the paper. (I.e., avoiding \(y\) as a probability, fixing the inequalities in the proof of Lemma 6, and introducing notation for function projection/restriction.) Thank you!

---

### Official Review · Reviewer_bM9f · 2024-11-10
**Review of "Understanding Aggregations of Proper Learners in Multiclass Classification"**

**Rating:** 7
**Confidence:** 3

**Review:**

This paper studies the sample complexity of aggregations of proper learners in multiclass learning. The authors proved that for hypothesis classes with finite graph dimension $d_G$, the majority vote of ERMs on subsets selected by some algorithms achieves a sample complexity of $O((d_G+\log(1/\delta))/\epsilon)$, which matches the lower bound they show for any majority vote of ERMs on certain classes of graph dimension $d_G$ and DS dimension 1. They also improve the lower bound of a single ERM on such classes. Finally, they prove the existence of hypotheses classes of DS dimension 1 for which any learner that achieves error strictly below 1/2 cannot be expressed as any function of finitely many proper learners.

Pros: The paper is organized and presented clearly. The problem studied is important in multiclass learning. The results are original with considerable contribution to the understanding of proper learners in multiclass learning.

Cons: I think the results and analysis for the two lower bounds (sec. 3.2 and sec. 4) are more interesting and significant. It would be better to move the lower bound sections to the front of the upper bound section in the first 12 pages. The proof for the upper bound is quite standard, so I believe it can be moved to the end of the paper so that the first 12 pages can be used to present the derivation of the lower bounds and the concept of properness numbers.

Is there a typo in Lemma 6? I think the RHS of the inequality should have $\bar{\mathcal A}$ instead of $\mathcal A$.

**Paper Award:**

No

---

> ### Author Response · Authors · 2024-11-22
>
> We thank the reviewer for their detailed feedback and careful reading of the paper.
>
> We agree with the reviewer in personally finding the analysis of the lower bounds more interesting than the upper bounds. Nevertheless, we feel the upper bounds may help serve as a warm-up for readers with less experience in the area, and that it may be favorable to structure the paper so as to begin with the upper bounds and conclude with the lower bounds. Nevertheless, we will take this feedback into careful consideration for the next version of the paper.
>
> Yes, that is a typo in Lemma 6 – thank you!

---

### Official Review · Reviewer_VuFr · 2024-11-16
**well-written work with solid results on multiclass PAC learning**

**Rating:** 8
**Confidence:** 3

**Review:**

This work studies the performance of majorities of ERM/proper learners in multiclass classification. Such learning algorithms have been shown to achieve better errors in the binary case. This work extends these results to the multiclass case via the standard reduction of multiclass to the binary case. The reduction introduces a dependence on the so called graph dimension of the concept class. The second question addressed by this work is whether such combinations of proper learners can be used to learn PAC learn an arbitrary learnable multiclass class. Here the result is negative. Showing that the dependence on the graph dimension is unavoidable. The result builds on a known construction and improves it by a log factor.
A significant plus is that the paper is written in a clear and detailed way making it accessible even to those who do not follow the area closely.

Overall this is a solid set of results that skillfully combines recent interesting developments in this area and enhances the understanding of multiclass PAC learning. A weaker side to work is that the setting of (multiclass) PAC model is rather detached from the motivation of multiclass ML being common in modern ML mentioned in the introduction. But that is not the fault of this particular work.

Minor comments:
p5: it looks like the function defines Plurality and not Majority
p.13 typo: "idenity"

**Paper Award:**

No

---

> ### Author Response · Authors · 2024-11-22
>
> We thank the reviewer for their detailed feedback and careful reading of the paper.
>
> The reviewer is indeed correct – the majority function we define is, strictly speaking, a pointwise plurality among the classifiers $f_i$ rather than a majority. (As we study general multiclass classification, a pointwise majority of classifiers is not guaranteed to exist, unlike in the binary case.) We will clarify this point in the next version of the paper.
>
> Thank you for catching the typo on page 13 as well!

---

### Meta-Review · Area_Chair_kaM5 · 2024-12-06

**Recommendation:** Accept
**Confidence:** 5

**Metareview:**

This paper provides a nice addition to the theoretical understanding of muticlass learning using aggregations of proper learners, including both lower bounds and upper bounds. The paper is technically sound and well written, and provides relevant contributions that are suitable for ALT.

**Paper Award:**

No